# Unbiased Attribution with Intrinsic Information

## Abstract

The importance of attribution algorithms in the AI field lies in enhancing model transparency, diagnosing and improving models, ensuring fairness, and increasing user understanding. Gradient-based attribution methods have become the most critical because of their high computational efficiency, continuity, wide applicability, and flexibility. However, current gradient-based attribution algorithms require the introduction of additional class information to interpret model decisions, which can lead to issues of information ignorance and extra information. Information ignorance can obscure important features relevant to the current model decision, while extra information introduces irrelevant data that can cause feature leakage in the attribution process. To address these issues, we propose the Attribution with Intrinsic Information (AII) algorithm, which analyzes model decisions without the need for specified class information. Additionally, to better evaluate the potential of current attribution algorithms, we introduce the metrics of insertion confusion and deletion confusion alongside existing mainstream metrics. To continuously advance research in the field of explainable AI (XAI), our algorithm is open-sourced at `https://anonymous.4open.science/r/AII-787D/`.

## 1 Introduction

As deep learning continues to advance rapidly, its performance in tasks such as image recognition (Xu et al., 2023; Liu et al., 2023) has reached unprecedented heights. These technological breakthroughs have brought revolutionary changes to various fields, including healthcare (Suganyadevi et al., 2022), autonomous driving (Grigorescu et al., 2020), and management decision-making (Shrestha et al., 2021). However, as these fields increasingly rely on deep learning technologies, the need for decision transparency has become more critical. If a model's decision-making process is not explainable, users may find it difficult to fully trust the results and to assign responsibility in the event of an incident.

Therefore, the research and development of Explainable AI (XAI) are of paramount importance. There has been extensive research in the XAI domain, with early interpretability methods such as Grad-CAM (Selvaraju et al., 2017) and LIME (Ribeiro et al., 2016) using heatmaps and local linear models to explain the decisions of Deep Neural Networks (DNNs). However, these methods have limitations in providing fine-grained and one-to-one explanations for each input feature. Consequently, researchers have proposed more detailed attribution methods, with Integrated Gradients (IG)Sundararajan et al. (2017) being one of the most significant. IG addresses the shortcomings of earlier methods and introduces axioms for attribution, providing a consistent and fair framework for feature importance. As research progressed, new adversarial example-based attribution methods were proposed, such as Adversarial Gradient Integration (AGI)Pan et al. (2021), MFABA (Zhu et al., 2024), and AttExplore (Zhu et al., 2023).

We note that existing attribution methods typically select specific class outputs or cross-entropy as the loss function and use backpropagation to obtain gradients concerning input samples to guide the attribution algorithm. We have identified two phenomena that will cause attribution bias due to such gradient selection: information ignorance and extra information. **Information ignorance refers to the omission of important features from classes not directly related to the model's final decision, while extra information involves the incorrect identification of irrelevant features**

**as significant.** Information ignorance leads to interpretability methods overlooking many features crucial to the model's current decision and failing to explain low-confidence situations (applicable under any non-100% confidence conditions). Extra information results in feature leakage (Shah et al., 2021), where features not contributing to the model's decision are incorrectly identified as important. To address these phenomena, we propose the Attribution with Intrinsic Information (AII) algorithm. In AII, we redefine the form of accumulated gradients and eliminate the need to introduce class information into the gradients. Additionally, we conduct rigorous mathematical derivations to ensure the validity of the AII algorithm and its adherence to attribution axioms (Sundararajan et al., 2017).

Beyond the extra information phenomenon introduced by attribution algorithms, it can also occur during the evaluation of attribution algorithms. This happens because neural networks cannot distinguish between feature removal behavior and the feature representation of black information. For image tasks, removing features and replacing their values with zero might be interpreted by the neural network as introducing black information, which is unfair for tasks where black is a key feature (e.g., black-and-white cat classification). Thus, we propose the fair insertion and fair deletion metrics to avoid bias during the evaluation process. Additionally, to assess the impact of confusion, we introduce the KL insertion and KL deletion metrics. We summarize our contributions as follows:

- We systematically pinpoint two phenomena that cause bias in current gradient-based attribution algorithms: information ignorance and extra information, which severely undermine the reliability of interpretability algorithms.

- To address the extra information phenomenon in the evaluation of attribution algorithms, we propose two fairer evaluation metrics.

- Based on the first contribution listed above, we design a novel gradient accumulation method and propose the AII algorithm, supported by rigorous mathematical derivations to ensure its stability.

- We open-source our experimental code to facilitate subsequent research and replication of experiments.

## 2 RELATED WORK

Methods for explaining deep neural networks (DNNs) have evolved through three distinct phases: local approximation and early layer-wise relevance propagation (LRP) methods (Bach et al., 2015), gradient-based attribution methods, and adversarial example-based attribution methods. Local approximation methods, such as LIME (Ribeiro et al., 2016), construct an approximate, more interpretable model in the vicinity of specific inputs to understand the behavior of the original model. LIME was the first to provide local interpretability using multiple interpretable structures near the sample, although it is time-consuming and the reliance on assumptions may be inaccurate. LRP, an early layer-wise relevance propagation method, analyzes sample features using the relevance of inputs, while DeepLIFT (Shrikumar et al., 2017), a general form of LRP, quantifies feature importance by comparing input features to predefined reference points, though it is highly sensitive to the choice of reference points. DeepLIFT does not satisfy the Implementation Invariance axiom proposed in Integrated Gradients (IG) (Sundararajan et al., 2017), leading to different attribution results for models with the same functionality. The limitations of these early methods have been discussed in (Zhu et al., 2024; 2023).

Gradient-based attribution methods leverage the gradient information during the training of neural networks to explain model decisions. These methods include Saliency Map (SM) (Simonyan et al., 2013), Grad-CAM (Selvaraju et al., 2017), Score-CAM (Wang et al., 2020), IG (Sundararajan et al., 2017), Fast IG (FIG) (Hesse et al., 2021), Expected Gradients (EG) (Erion et al., 2021), Smooth-Grad (SG) (Smilkov et al., 2017), and Guided IG (GIG) (Kapishnikov et al., 2021). SM computes the gradient of input features concerning the model output to identify the most important features, but it is prone to gradient saturation issues. Grad-CAM and Score-CAM use gradient information from intermediate layers for explanations, though they cannot provide high-resolution, fine-grained explanations. IG addresses the limitations of SM by integrating gradients along a path from the baseline to the input, but it has a high computational cost, which Fast IG mitigates by improving numerical integration techniques to accelerate IG. EG enhances stability and consistency by averag-

ing gradients over multiple baselines, while SG improves the smoothness and stability of attribution results by adding random noise to the input, though it may obscure important subtle features. GIG combines the principles of IG and guided backpropagation to selectively backpropagate gradients, enhancing interpretability but potentially overemphasizing directly related features while neglecting equally important indirect features.

Adversarial example-based attribution methods represent a highly effective branch of gradient-based attribution methods. These methods provide deeper explanations by generating adversarial examples and exploring model decision boundaries, thus avoiding the need to manually specify baseline points. However, this class of methods introduces numerous intermediate states from out-of-distribution (OOD) space during adversarial attacks, leading to the introduction of extra information into the attribution process. Representative methods in this category include Adversarial Gradient Integration (AGI) (Pan et al., 2021), Boundary-based Integrated Gradients (BIG) (Wang et al., 2021), AttEXplore (Zhu et al., 2023), and More Faithful and Accelerated Boundary-based Attribution (MFABA) (Zhu et al., 2024). AGI uses targeted adversarial attacks to explore decision boundaries and improves attribution performance through non-linear path-integrated gradients. BIG introduces boundary search mechanisms to optimize baseline selection, achieving more accurate feature attribution, but the linear integration path limits its ability to capture the non-linearity and complexity of model decisions. MFABA improves explanation accuracy and computational efficiency through second-order Taylor expansion and decision boundary exploration. AttEXplore (Zhu et al., 2023) combines adversarial attacks with model parameter exploration, emphasizing transition capabilities across different decision boundaries, thus enhancing the generalizability of its interpretability.

It is noteworthy that these gradient-based attribution methods typically use the output of the maximum class or cross-entropy as the loss function during gradient accumulation. This leads to the information ignorance and extra information phenomena discussed in the following sections, which significantly exacerbate the bias of attribution algorithms. We will analyze these issues in detail in Section 3.2.

## 3 METHOD

In this section, we first define the problem, then introduce the phenomena of "**Information Ignorance**" and "**Extra Information**" in current attribution algorithms, analyze why Extra Information leads to feature leakage problems, and propose new evaluation methods to address Extra Information in the evaluation of attribution algorithms. Finally, we present the AII algorithm, which can avoid the occurrence of bias.

### 3.1 PROBLEM DEFINITION

Given the neural network parameters $w \in \mathbb{R}^m$ and the sample to be attributed $x \in \mathbb{R}^n$, our goal is to use attribution methods to obtain the attribution result $A(x) \in \mathbb{R}^n$, where $A_i(x)$ represents the importance of the $i$-th feature dimension, $n$ represents the number of dimensions. The larger the attribution result, the more important that dimension is for the model's decision. We use $f_j(x) \in \mathbb{R}$ to represent the model's output for the $j$-th class, and $P_j(x)$ to denote the probability of the $j$-th class after applying the softmax function.

### 3.2 INFORMATION IGNORANCE AND EXTRA INFORMATION

In current gradient-based attribution methods, $\frac{\partial L(f(x),y)}{\partial x}$ is typically chosen as the gradient, where $y = \arg\max_j f_j(x)$, and $L$ is usually the negative of the output value of the class with the maximum output or the cross-entropy loss function. Intuitively, we use the class with the highest output from the model for the input sample $x$ to guide the attribution algorithm. This introduces "Extra Information," which is detrimental to the attribution algorithm. Additionally, since the loss function $L(f(x), y)$ contains only the class information of $y$, this leads to the phenomenon of Information Ignorance.

**Information Ignorance**: Information Ignorance refers to the fact that attribution methods tend to ignore the feature information of classes other than the target class. For example, as shown in Figure 1, when the target class is the dog, attribution methods ignore the features of the cat. Conversely,

when the target class is the cat, the dog's features are ignored. However, in reality, the model considers both the dog and the cat for making decisions. For instance, when the confidence for the dog class is 0.53, the model attends to both the dog's and the cat's features. This indicates that during the decision-making process, the model uses information from both the dog and the cat features.

$$L(f(x + \Delta x), y) \approx L(f(x), y) + \Delta x^\top \cdot \frac{\partial L(f(x), y)}{\partial x} + \mathcal{O} \tag{1}$$

We first gain an intuitive understanding of the Information Ignorance phenomenon from the definition of the gradient. By performing a first-order Taylor expansion of the loss function as in Equation 1, where $\mathcal{O}$ represents higher-order infinitesimals (which are ignored in the first-order analysis), we can observe the sensitivity of different dimensions in $x$ to the loss function $L(f(x), y)$. Therefore, SM (Simonyan et al., 2013) directly uses $\frac{\partial L(f(x), y)}{\partial x}$ as the interpretability result. Accumulating $\frac{\partial L(f(x), y)}{\partial x}$ during the change process of the sample $x$ reveals the overall performance of sensitivity, which is the idea behind methods such as AGI (Pan et al., 2021), BIG (Wang et al., 2021), MFABA (Zhu et al., 2024), and AttExplore (Zhu et al., 2023). However, it is important to note that $\frac{\partial L(f(x), y)}{\partial x}$ is only responsible for the loss function $L(f(x), y)$, which in turn is only responsible for the class $y$. This means that attribution methods using $\frac{\partial L(f(x), y)}{\partial x}$ will ignore information from classes other than the specified class $y$.

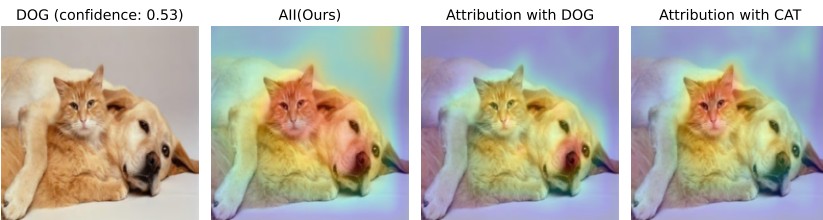

Figure 1: Illustration of the **Information Ignorance** phenomenon. Attribution results using the AttExplore (Zhu et al., 2023) algorithm on an image containing both a cat and a dog. The model's classification confidence is 0.53, indicating that the model considers both the cat and the dog when making its decision. However, traditional attribution methods only focus on the predefined label, in this case, the dog, and ignore the cat's features, leading to the Information Ignorance phenomenon. In contrast, our method, which does not rely on predefined labels, is able to attribute the features of both the cat and the dog, avoiding the Information Ignorance issue.

As shown in Figure 1, the AttExplore (Zhu et al., 2023) algorithm only provides information about the dog class, although the model also considers the cat area in its decision-making process. The Information Ignorance phenomenon worsens when the confidence of the current class is low. In other words, **current gradient-based attribution methods cannot explain why the model has low confidence in its decision** (e.g., the presence of a cat explains why the dog class confidence is only 0.53). The AII algorithm, introduced below, avoids Information Ignorance by not using class information as guidance. More experimental results on AII for low-confidence data can be found in Section 4.5.

Next, we analyze the causes of the Extra Information phenomenon. In the process of specifying the loss function class, previous attribution algorithms use the output of the class with the maximum output, introducing Extra Information. Intuitively, we impose the concept that the input sample belongs to the current class onto the interpretability method.

**Extra Information**: Extra Information refers to the introduction of irrelevant features into the image that are not related to the current task, which attribution methods still focus on. For example, as shown in Figure 2, attribution methods focus on irrelevant extra features that do not belong to any class in the original image. The square region at the bottom of the original image is not part of any class, but if we force the attribution method to focus on class 0, even with extremely low confidence, we can see that some attribution methods will focus on these extra irrelevant features. This additional information results in attribution bias.

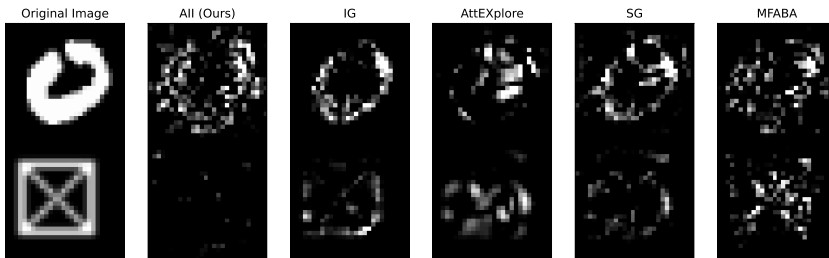

Figure 2: Illustration of the **Extra Information** phenomenon. As shown in the image, ⊠ represents extra information that does not belong to any class feature. The model does not attend to this region, yet other attribution methods, apart from ours, display that the model focuses on the ⊠ feature. This demonstrates that our method can effectively avoid the Extra Information phenomenon.

For example, the feature leakage problem mentioned in (Shah et al., 2021) is a manifestation of Extra Information. As shown in Figure 2, the grid patterns do not contribute to the handwritten digit recognition task, but in the interpretability process, under extremely low confidence, we regard them as part of one of the ten classes and interpret them. This additional information causes attribution bias. In other words, the information "leaks" because we introduce redundant information.

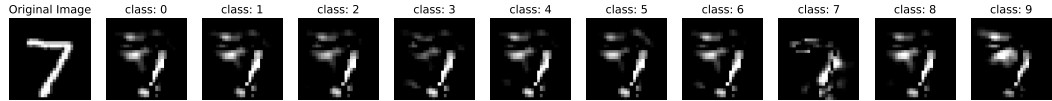

Figure 3: Attribution results for digit recognition. No significant class differences observed when changing the accumulated gradient class.

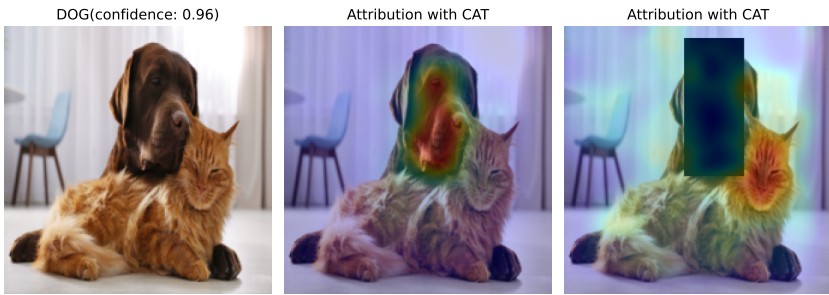

Figure 4: Attribution results using the cat class on a dog image. Important features mistakenly identified due to model training errors.

We also explore the interesting viewpoint that specifying the class might help identify important features for a specific class. However, as shown in Figure 3, using AttExplore and modifying the accumulated gradient class for digits 0-9 does not show significant class differences for digit 7. Similarly, Figure 4 shows that using the cat class for attribution on a dog image highlights features on the dog. This could indicate a model training error, where the dog's features are mistaken for the cat's core features. Masking the core feature area shifts the important attribution region to the cat, indicating the model can respond to the correct cat features. We suspect this is because, during the accumulated gradient process, the changes in lower confidence classes are limited (Zhu et al., 2023), making the class specification process not always effective.

**Remark 1.** *Specifying class information in the gradient calculation during attribution can lead to information loss and the phenomenon of Extra Information.*

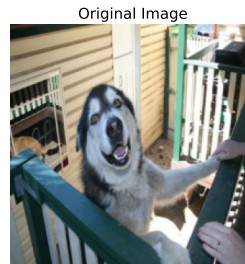 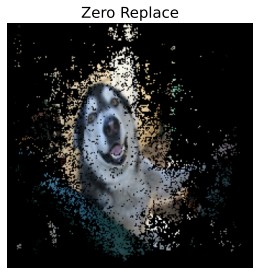 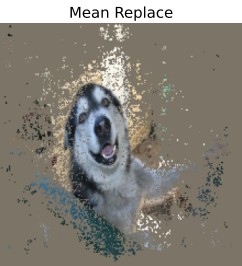

Original Image · Zero Replace · Mean Replace

Figure 5: Comparison of Different Feature Replacement Methods

### 3.3 EXTRA INFORMATION IN ATTRIBUTION EVALUATION

Besides the Extra Information phenomenon in the attribution process, it also appears in the evaluation of attribution algorithms. The core idea of evaluating attribution algorithms is to insert information from the original image into an all-black pixel image based on the attribution score from high to low, assessing how quickly the current model decision can be recovered, corresponding to the Insertion score. Conversely, the Deletion score replaces the original image pixels with black pixels from high to low based on the attribution score, evaluating how quickly the current model decision can be destroyed. However, this process inherently assumes that a black image represents "no information." In reality, **the model cannot distinguish between feature removal and features represented by black pixels.** For instance, in the task of distinguishing between black and white cats, the color of the cat's fur is an important feature. Zeroing out the features during this process makes a white cat more likely to be seen as a black cat, introducing black cat features rather than removing features.

To replace the zero-out operation with a more reasonable feature removal behavior, we designed the **Confusion Feature Algorithm (CFA)**.

$$x^* = \max_x \mathcal{H}(x) = \sum_{i=1}^{C} P_j(x) \log P_j(x) \quad \text{s.t.} \quad x_1 = x_2 = x_3 \cdots = x_n \tag{2}$$

The core idea of CFA is to find the pixel value that maximizes the entropy of the model output distribution (the higher the entropy, the greater the uncertainty of the model decision) and ensure that the entire input consists of the same pixel value, aligning with human intuition of feature removal by replacing large areas with the same pixel, as shown in Figure 5. This approach makes the explanation process more consistent with human understanding. Because a single pixel value is easier for humans to comprehend, interpretability methods should aim for results that closely approximate human intuition.

Since finding the optimal solution for Equation 2 is difficult, we use gradient descent to iteratively find the optimal solution. Note that all $x^t$ are the same, and the features input into the network are obtained by repeating one pixel. The proof is provided in Appendix B.

$$x^t = x^{t-1} - \alpha \cdot \text{sign}\left( \frac{\partial \sum_{j=1}^{C} \log P_j(r(x^{t-1}))}{\partial x} \right) \tag{3}$$

Here, $x^0 \sim U(0,1)$, $U$ represents a uniform distribution (this operation maps pixel values to 0-1 through normalization). $x^0 \in \mathbb{R}^3$ for image tasks, representing the RGB values of one pixel. $r$ is a repeat function that repeats the pixel to match the input dimensions of the neural network. To avoid local optima, we sample multiple times and select the $x$ that maximizes $\mathcal{H}(x)$ as the final choice. Replacing the all-black image in the attribution evaluation process with the learned $x$ avoids the Extra Information phenomenon mentioned above. We name the replaced algorithms as Fair Insertion and Fair Deletion metrics. Additionally, to evaluate the model's uncertainty in the attribution process, we propose KL Insertion and KL Deletion metrics, assessing the change in model decision confusion during the attribution insertion process. Additionally, Our AII method has complexity comparable to AGI (Pan et al., 2021). Each time we compute Equation 3, we perform both a forward pass and a backward pass. This means that during the update process, there will be $m \times t$ forward and backward propagations, where $m$ is the number of samples and $t$ is the number of iterations.

## 3.4 ATTRIBUTION WITH INTRINSIC INFORMATION (AII)

$$A(x) = \int \Delta x^t \cdot \frac{\partial \sum_{j=1}^{c} \log P_j(x^t)}{\partial x^t} dt \tag{4}$$

Equation 4 represents the core formula of the AII algorithm. In the attribution process, we design the gradient to be accumulated as $\frac{\partial \sum_{j=1}^{c} \log P_j(x^t)}{\partial x^t}$, avoiding the phenomena of Information Ignorance and Extra Information. $\Delta x^t$ follows the targeted adversarial attack update strategy from AGI (Pan et al., 2021). Note that the only constraint on $\Delta x^t$ is $|\Delta x^t| \leq \epsilon$, where $\epsilon$ is typically limited to one pixel. Since the importance of each feature dimension is calculated independently, for the i-th dimension, the feature importance can be expressed as $A(x_i) = \int \Delta x_i^t \cdot \frac{\partial \sum_{j=1}^{c} \log P_j(x_i^t)}{\partial x_i^t} dt$. Additionally, the AII algorithm satisfies the Sensitivity axiom and Implementation Invariance axiom, with the proof in Appendix C. Detailed pseudocode will be provided in Appendix D.

**Remark 2.** *Any feature changes that increase model decision uncertainty can be captured by Equation 4.*

## 4 EXPERIMENTS

In this section, we provide a detailed account of the experiments conducted to evaluate the AII method. This includes information on the datasets used, models, baseline methods, evaluation metrics, and an analysis of the results.

### 4.1 DATASET AND MODELS

Following previous work such as AGI (Pan et al., 2021), MFABA (Zhu et al., 2024), and AttExplore (Zhu et al., 2023), we randomly selected 1000 images from the ImageNet dataset (Deng et al., 2009) to maintain the consistency of the experiment. For our models, we chose three classic deep learning architectures: Inception-v3 (Szegedy et al., 2016), ResNet-50 (He et al., 2016), and VGG16 (Simonyan & Zisserman, 2014).

### 4.2 BASELINES

To comprehensively evaluate and compare our method, we selected 11 existing interpretability methods. The criteria for selection included publication in top-tier academic conferences and the availability of open-source code. The methods compared are AttEXplore (Zhu et al., 2023), AGI (Pan et al., 2021), MFABA (Zhu et al., 2024), BIG (Wang et al., 2021), EG (Erion et al., 2021), FIG (Hesse et al., 2021), DeepLIFT (Shrikumar et al., 2017), SG (Smilkov et al., 2017), SM (Simonyan et al., 2013), GIG (Kapishnikov et al., 2021), and IG (Sundararajan et al., 2017).

### 4.3 PARAMETERS

In all experiments, we used two NVIDIA A100 40GB GPUs. Our method involves two hyperparameters: the number of explorations $M$ and the number of attack iterations $T$, both set to 20. More ablation experiment results are provided in the Appendix E.6.

### 4.4 EVALUATION METRICS

We used four groups of evaluation metrics to assess the performance of the attribution methods.

The Insertion Score (INS) evaluates the area under the curve (AUC) as information from the original image is incrementally inserted into an all-black pixel image based on the attribution score, from high to low (Samek et al., 2016). This metric measures how quickly the current model decision can be recovered. The Deletion (DEL) Score evaluates the AUC as pixels from the original image are progressively replaced with black pixels based on the attribution score, from high to low. This metric assesses how quickly the current model decision can be disrupted. **Compared to the DEL, the INS is generally considered more important** (Pan et al., 2021). Therefore, to eliminate the inconsistency between these two parameters in this paper, we use the GAP metric, which is the

difference between INS and DEL, for unified comparison. The results of INS and DEL are provided in Appendix E.1.

We found discrepancies in the implementation of Insertion Score and Deletion Score in the open-source code of RISE (Petsiuk et al., 2018), MFABA (Zhu et al., 2024), BIG (Wang et al., 2021), and AGI (Pan et al., 2021). Previous work sorted the importance of each image channel separately, whereas the evaluation process should treat each input dimension equally. Therefore, we improved this by not calculating separately for each channel. We refer to these improved metrics as the Unified Insertion Score (U-INS) and Unified Deletion Score (U-DEL).

To address the issue of extra information in the evaluation process, we propose the Fair Insertion and Fair Deletion metrics. The Fair Insertion (F-INS) metric replaces the initial all-black image in the Insertion score with the learned $x^*$. Similarly, the Fair Deletion (F-DEL) metric replaces the zero-out operation in the Deletion score with the $x^*$ pixel operation. These modifications aim to provide a more accurate assessment by avoiding the introduction of extra information.

Furthermore, we introduce the KL Insertion (KL-INS) and KL Deletion (KL-DEL) metrics to evaluate the change in model decision uncertainty during the attribution process. KL Insertion replaces the current class output probability with the KL divergence $KL(Q, P(x))$ and calculates the AUC of the KL curve. A smaller area indicates that the important features identified by the attribution method quickly reduce the model's decision uncertainty. Conversely, KL Deletion calculates the AUC of the KL curve as important features are progressively removed, with a larger value indicating a rapid increase in model decision uncertainty. To illustrate this, we use the case where entropy is maximized. For example, in a scenario with 1000 classes, if the model's output probability for each class is exactly $\frac{1}{1000}$, the model's uncertainty reaches its maximum. This serves as a reference point for measuring the reduction in uncertainty during the attribution process.

## 4.5 RESULTS

In this section, we provide additional experiments in Tables 1, 2, and 3, where we further analyze the performance of various attribution methods on datasets split by confidence levels ($<70\%$ and $\geq 70\%$). From these results, it is evident that our proposed method consistently achieves the best performance across both high and low confidence datasets. This demonstrates that our method is less susceptible to the effects of Information Ignorance and Extra Information, making it more robust compared to other methods. More experiments, such as the performance on traditional attribution metrics like Insertion and Deletion, results on the full dataset without confidence level distinctions, performance on Transformer-based models, and ablation studies can be found in Appendix E.

Table 1: Evaluation of various interpretability methods via U-INS and U-DEL metrics. ↑ indicates that higher values in the column correspond to better interpretability performance, while ↓ indicates that lower values correspond to better interpretability performance. The * symbol denotes the primary reference metric for comparison. The table is divided into three confidence-based categories: Low Confidence ($<70\%$), High Confidence ($\geq 70\%$).

| | Inception-v3 | | | | | | ResNet-50 | | | | | | VGG16 | | | | | |
| | Low Confidence ($<70\%$) | | | High Confidence ($\geq 70\%$) | | | Low Confidence ($<70\%$) | | | High Confidence ($\geq 70\%$) | | | Low Confidence ($<70\%$) | | | High Confidence ($\geq 70\%$) | | |
| Method | U-INS (↑) | U-DEL (↓) | GAP* (↑) | U-INS (↑) | U-DEL (↓) | GAP* (↑) | U-INS (↑) | U-DEL (↓) | GAP* (↑) | U-INS (↑) | U-DEL (↓) | GAP* (↑) | U-INS (↑) | U-DEL (↓) | GAP* (↑) | U-INS (↑) | U-DEL (↓) | GAP* (↑) |
|---|---|---|---|---|---|---|---|---|---|---|---|---|---|---|---|---|---|---|
| SM | 0.0226 | 0.0349 | -0.0123 | 0.0889 | 0.047 | 0.0419 | 0.0283 | 0.015 | 0.0133 | 0.0658 | 0.0369 | 0.0289 | 0.0216 | 0.0137 | 0.0079 | 0.054 | 0.0228 | 0.0312 |
| IG | 0.0228 | 0.0283 | -0.0055 | 0.1009 | 0.0312 | 0.0697 | 0.0228 | 0.0108 | 0.012 | 0.0519 | 0.0254 | 0.0265 | 0.0164 | 0.0103 | 0.0061 | 0.0392 | 0.0176 | 0.0216 |
| FIG | 0.028 | 0.019 | 0.009 | 0.0467 | 0.0744 | -0.0277 | 0.0119 | 0.0211 | -0.0092 | 0.0327 | 0.0466 | -0.0139 | 0.0119 | 0.0157 | -0.0038 | 0.0218 | 0.0356 | -0.0138 |
| BIG | 0.0904 | 0.0328 | 0.0576 | 0.1843 | 0.0844 | 0.0999 | 0.047 | 0.0294 | 0.0176 | 0.118 | 0.0693 | 0.0487 | 0.0331 | 0.021 | 0.0121 | 0.088 | 0.0501 | 0.0379 |
| MFABA | 0.0975 | 0.0331 | 0.0644 | 0.2799 | 0.0859 | 0.194 | 0.0504 | 0.0305 | 0.0199 | 0.1401 | 0.078 | 0.0621 | 0.0369 | 0.022 | 0.0149 | 0.1133 | 0.0529 | 0.0604 |
| AttEXplore | 0.1324 | 0.0321 | 0.1003 | 0.3757 | 0.0739 | 0.3018 | 0.0992 | 0.0226 | 0.0766 | 0.2745 | 0.053 | 0.2215 | 0.0807 | 0.0205 | 0.0602 | 0.2444 | 0.0468 | 0.1976 |
| GIG | 0.0242 | 0.0234 | 0.0008 | 0.0992 | 0.0287 | 0.0705 | 0.0225 | 0.0103 | 0.0122 | 0.0517 | 0.0183 | 0.0334 | 0.0187 | 0.0094 | 0.0093 | 0.0409 | 0.0128 | 0.0281 |
| EG | 0.1697 | 0.1848 | -0.0151 | 0.398 | 0.4521 | -0.0541 | 0.1162 | 0.1243 | -0.0081 | 0.2988 | 0.3465 | -0.0477 | 0.0924 | 0.0772 | 0.0152 | 0.2217 | 0.1972 | 0.0245 |
| DeepLIFT | 0.0269 | 0.0209 | 0.006 | 0.0811 | 0.0573 | 0.0238 | 0.0189 | 0.0121 | 0.0068 | 0.0445 | 0.0333 | 0.0112 | 0.0165 | 0.0105 | 0.006 | 0.0355 | 0.0198 | 0.0157 |
| SG | 0.0423 | 0.0299 | 0.0124 | 0.1957 | 0.0251 | 0.1706 | 0.0663 | 0.0105 | 0.0558 | 0.1317 | 0.0175 | 0.1142 | 0.0545 | 0.0098 | 0.0447 | 0.1296 | 0.0134 | 0.1162 |
| AGI | 0.1104 | 0.029 | 0.0814 | 0.374 | 0.084 | 0.29 | 0.0905 | 0.0262 | 0.0643 | 0.3684 | 0.0654 | 0.303 | 0.0537 | 0.0189 | 0.0348 | 0.2935 | 0.0475 | 0.246 |
| AII (Ours) | 0.2119 | 0.0385 | **0.1734** | 0.5131 | 0.1056 | **0.4075** | 0.1436 | 0.031 | **0.1126** | 0.3859 | 0.0678 | **0.3181** | 0.1026 | 0.0219 | **0.0807** | 0.3118 | 0.0492 | **0.2626** |

**U-INS and U-DEL:** In this section, we analyze the performance of our method AII compared to various state-of-the-art attribution methods using the Unified Insertion Score (U-INS) and Unified Deletion Score (U-DEL). These metrics provide a more unified and consistent evaluation by treating each input dimension equally, unlike previous implementations.

Table 2: Evaluation of various interpretability methods via U-INS and U-DEL metrics.

| | Inception-v3 | | | | | | ResNet-50 | | | | | | VGG16 | | | | | |
|---|---|---|---|---|---|---|---|---|---|---|---|---|---|---|---|---|---|---|
| | Low Confidence (<70%) | | | High Confidence (≥70%) | | | Low Confidence (<70%) | | | High Confidece (≥70%) | | | Low Confidence (<70%) | | | High Confidence (≥70%) | | |
| Method | F-INS (↑) | F-DEL (↓) | GAP* (↑) | F-INS (↑) | F-DEL (↓) | GAP* (↑) | F-INS (↑) | F-DEL (↓) | GAP* (↑) | F-INS (↑) | F-DEL (↓) | GAP* (↑) | F-INS (↑) | F-DEL (↓) | GAP* (↑) | F-INS (↑) | F-DEL (↓) | GAP* (↑) |
| SM | 0.024 | 0.0207 | 0.0033 | 0.0549 | 0.0634 | -0.0085 | 0.0187 | 0.0294 | -0.0107 | 0.047 | 0.068 | -0.021 | 0.0133 | 0.0188 | -0.0055 | 0.0238 | 0.0408 | -0.017 |
| IG | 0.0306 | 0.0288 | 0.0018 | 0.0586 | 0.0934 | -0.0348 | 0.0222 | 0.0601 | -0.0379 | 0.0527 | 0.1305 | -0.0778 | 0.0172 | 0.0343 | -0.0171 | 0.0282 | 0.0788 | -0.0506 |
| FIG | 0.0276 | 0.0309 | -0.0033 | 0.0845 | 0.0746 | 0.0099 | 0.0561 | 0.0271 | 0.029 | 0.1188 | 0.0671 | 0.0517 | 0.0319 | 0.0205 | 0.0114 | 0.0729 | 0.033 | 0.0399 |
| BIG | 0.0802 | 0.0296 | 0.0506 | 0.1629 | 0.0648 | 0.0981 | 0.0488 | 0.035 | 0.0138 | 0.1399 | 0.0776 | 0.0623 | 0.0331 | 0.019 | 0.0141 | 0.0729 | 0.0411 | 0.0318 |
| MFABA | 0.0923 | 0.0288 | 0.0635 | 0.2632 | 0.0679 | 0.1953 | 0.0533 | 0.037 | 0.0163 | 0.1601 | 0.0877 | 0.0724 | 0.0377 | 0.0195 | 0.0182 | 0.1018 | 0.0422 | 0.0596 |
| AtteXplore | 0.1262 | 0.0218 | 0.1044 | 0.3503 | 0.0462 | 0.3041 | 0.0943 | 0.0279 | 0.0664 | 0.2881 | 0.0603 | 0.2278 | 0.0756 | 0.0161 | 0.0595 | 0.215 | 0.036 | 0.179 |
| GIG | 0.0288 | 0.0284 | 0.0004 | 0.0615 | 0.0889 | -0.0274 | 0.0198 | 0.0539 | -0.0341 | 0.0485 | 0.1163 | -0.0678 | 0.0154 | 0.0309 | -0.0155 | 0.027 | 0.0708 | -0.0438 |
| EG | 0.1306 | 0.1384 | -0.0078 | 0.3073 | 0.3241 | -0.0168 | 0.0838 | 0.0951 | -0.0113 | 0.2282 | 0.2769 | -0.0487 | 0.0629 | 0.0612 | 0.0017 | 0.1643 | 0.1433 | 0.021 |
| DeepLIFT | 0.0301 | 0.0296 | 0.0005 | 0.0779 | 0.0879 | -0.01 | 0.0273 | 0.0576 | -0.0303 | 0.0748 | 0.1182 | -0.0434 | 0.0157 | 0.0363 | -0.0206 | 0.031 | 0.0742 | -0.0432 |
| SG | 0.0241 | 0.0255 | -0.0014 | 0.0408 | 0.0779 | -0.0371 | 0.0124 | 0.0577 | -0.0453 | 0.0224 | 0.1156 | -0.0932 | 0.0096 | 0.0318 | -0.0222 | 0.0154 | 0.0781 | -0.0627 |
| AGI | 0.1211 | 0.0217 | 0.0994 | 0.3734 | 0.0555 | 0.3179 | 0.0808 | 0.026 | 0.0548 | 0.3784 | 0.0661 | 0.3123 | 0.0545 | 0.0157 | 0.0388 | 0.2538 | 0.039 | 0.2148 |
| AII (our) | 0.2139 | 0.0273 | **0.1866** | 0.5223 | 0.072 | **0.4503** | 0.1305 | 0.0368 | **0.0937** | 0.3881 | 0.0705 | **0.3176** | 0.0965 | **0.0164** | 0.0801 | 0.2738 | 0.0372 | **0.2366** |

Table 3: Evaluation of various interpretability methods via KL-INS and KL-DEL metrics.

| | Inception-v3 | | | | | | ResNet-50 | | | | | | VGG16 | | | | | |
|---|---|---|---|---|---|---|---|---|---|---|---|---|---|---|---|---|---|---|
| | Low Confidence (<70%) | | | High Confidence (≥70%) | | | Low Confidence (<70%) | | | High Confidence (≥70%) | | | Low Confidence (<70%) | | | High Confidence (≥70%) | | |
| Method | KL-INS (↑) | KL-DEL (↓) | GAP* (↑) | KL-INS (↑) | KL-DEL (↓) | GAP* (↑) | KL-INS (↑) | KL-DEL (↓) | GAP* (↑) | KL-INS (↑) | KL-DEL (↓) | GAP* (↑) | KL-INS (↑) | KL-DEL (↓) | GAP* (↑) | KL-INS (↑) | KL-DEL (↓) | GAP* (↑) |
| SM | 4.0006 | 4.1461 | -0.1455 | 4.2733 | 4.2992 | -0.0259 | 5.7474 | 5.7749 | -0.0275 | 5.8957 | 6.0216 | -0.1259 | 4.0809 | 4.0889 | -0.008 | 4.1862 | 4.3423 | -0.1561 |
| IG | 4.3213 | 4.478 | -0.1567 | 4.659 | 4.7538 | -0.0948 | 5.1251 | 5.2361 | -0.111 | 5.2658 | 5.8433 | -0.5775 | 3.9275 | 4.0115 | -0.084 | 4.1348 | 4.4998 | -0.365 |
| FIG | 4.3368 | 4.2119 | 0.1249 | 4.5255 | 4.5143 | 0.0112 | 5.1004 | 5.0536 | 0.0468 | 5.5885 | 5.2709 | 0.3176 | 3.8839 | 3.8245 | 0.0594 | 4.3227 | 3.9944 | 0.3283 |
| BIG | 5.7597 | 4.0682 | 1.6915 | 5.8104 | 4.2271 | 1.5833 | 5.5494 | 4.1986 | 1.3508 | 5.8397 | 4.7798 | 1.0599 | 4.2773 | 3.8223 | 0.455 | 4.5892 | 4.0533 | 0.5359 |
| MFABA | 5.7783 | 4.0013 | 1.777 | 7.1895 | 4.0598 | 3.1297 | 5.6028 | 4.075 | 1.5278 | 6.2243 | 4.3752 | 1.8491 | 4.454 | 3.8081 | 0.6459 | 4.8555 | 3.9935 | 0.862 |
| AtteXplore | 6.4325 | 3.4516 | 2.9809 | 8.3455 | 3.4832 | 4.8623 | 6.2566 | 5.0651 | 1.1915 | 7.2563 | 5.2456 | 2.0107 | 6.0355 | 4.1755 | 1.86 | 6.9415 | 4.2988 | 2.6427 |
| GIG | 4.3707 | 4.4534 | -0.0827 | 4.6371 | 4.6904 | -0.0533 | 5.3644 | 5.4411 | -0.0767 | 5.4456 | 5.9125 | -0.4669 | 3.9515 | 3.9936 | -0.0421 | 4.4372 | 4.2935 | -0.2563 |
| EG | 4.9925 | 4.8459 | 0.1466 | 6.615 | 5.9209 | 0.6941 | 4.9839 | 4.6738 | 0.3101 | 5.6367 | 5.3364 | 0.3003 | 5.0065 | 4.906 | 0.1005 | 5.8683 | 5.6151 | 0.2532 |
| DeepLIFT | 4.1727 | 4.2879 | -0.1152 | 4.484 | 4.4823 | 0.0017 | 4.8818 | 5.1143 | -0.2325 | 5.1481 | 5.5994 | -0.4513 | 3.7093 | 4.0851 | -0.3758 | 3.8975 | 4.4789 | -0.5814 |
| SG | 4.173 | 4.1349 | 0.0381 | 4.2128 | 4.254 | -0.0412 | 6.2011 | 6.1561 | 0.045 | 6.1479 | 6.7031 | -0.5552 | 4.7479 | 4.7833 | -0.0354 | 4.8112 | 5.0538 | -0.2426 |
| AGI | 5.5176 | 4.0858 | 1.4318 | 8.2868 | 4.078 | 4.2088 | 5.8622 | 4.5894 | 1.2728 | 7.5437 | 4.431 | 3.1127 | 5.5245 | 4.1857 | 1.0388 | 7.6075 | 4.489 | 3.1185 |
| AII (our) | 7.6129 | 3.6638 | **3.9491** | 10.802 | 3.789 | **7.013** | 6.3754 | 3.9806 | **2.3948** | 7.6201 | 4.2561 | **3.364** | 6.0499 | 4.4146 | **1.6353** | 7.9069 | 4.6668 | **3.2401** |

Table 1 summarizes the performance of various interpretability methods, including AII, evaluated on Inception-v3, ResNet-50, and VGG16 models using the U-INS and U-DEL metrics. Our method, AII, consistently outperforms other advanced attribution methods, achieving the highest GAP scores across all models. AII excels in both high-confidence and low-confidence datasets, providing superior interpretability by delivering more faithful explanations across different confidence levels. The average improvements of AII over other methods are as high as 0.2232, 0.2049, and 0.1421 on the three models, respectively. Specifically, on high-confidence data, AII achieves an average GAP improvement of 0.2466 over other methods, demonstrating its superior performance under normal conditions. Additionally, AII achieves an average improvement of 0.099 on low-confidence data, indicating its robustness in addressing the Information Ignorance phenomenon.

**F-INS and F-DEL:** In this section, we evaluate the performance of our AII method and other interpretability methods using the Fair Insertion (F-INS) and Fair Deletion (F-DEL) metrics. These metrics provide a fairer and more precise evaluation by mitigating the introduction of extra information.

Table 2 presents the performance results across the three models. Compared to other methods, AII demonstrates significant improvements, with an average increase of 0.1085 on low-confidence data. Specifically, AII improves performance by 0.1583 on Inception-v3, 0.0927 on ResNet-50, and 0.0744 on VGG16. On high-confidence data, AII achieves an average GAP improvement of 0.2896, including a performance increase of 0.3784 on Inception-v3, 0.2835 on ResNet-50, and 0.2067 on VGG16. These results highlight the consistent and substantial advantages of AII over other methods in both low- and high-confidence scenarios, further establishing its robustness and effectiveness in interpretability tasks.

**KL-INS and KL-DEL:** In this section, we analyze the performance of our AII method and other interpretability methods using the KL Insertion (KL-INS) and KL Deletion (KL-DEL) metrics. These metrics evaluate the change in model decision uncertainty during the attribution process, providing a comprehensive assessment of how quickly the model's decision uncertainty is reduced or increased by the identified important features.

Table 3 shows the results across the three models, and the performance of our method, AII, demonstrates even more significant advantages compared to other attribution methods under this evaluation metric. Specifically, on low-confidence data, our AII method achieved an average GAP improvement of 2.1566, with respective improvements of 3.2499 on Inception-v3, 1.9132 on ResNet-50, and

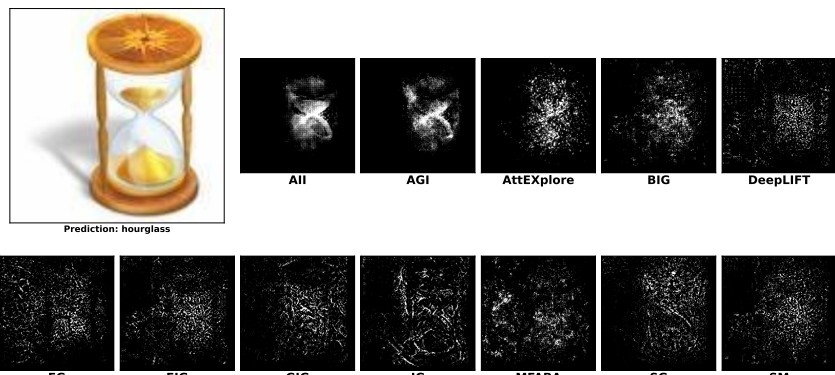

Figure 6: AII Attribution Result on Inception-v3

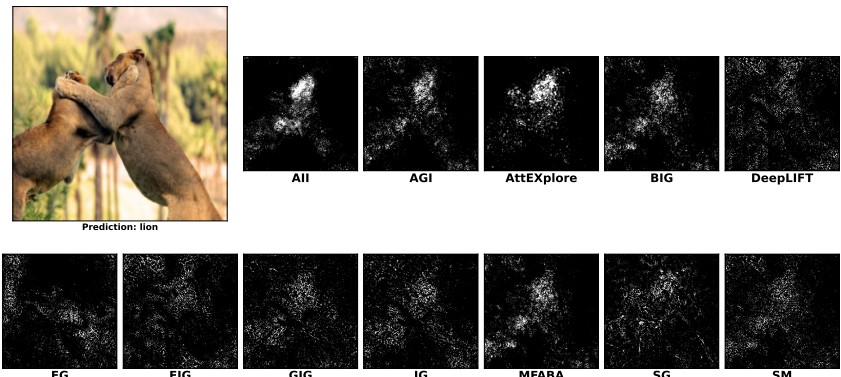

Figure 7: AII Attribution Result on VGG16

1.3067 on VGG16. On high-confidence data, AII achieved an even more pronounced average GAP improvement of 3.7242, with respective gains of 5.7152 on Inception-v3, 2.7755 on ResNet-50, and 2.6820 on VGG16. These results highlight the substantial and consistent superiority of AII across both confidence levels, solidifying its effectiveness and robustness in interpretability tasks across diverse models and datasets.

**Attribution Results:** As shown in Figures 6 and 7, the attribution results for both the VGG16 and Inception-v3 models demonstrate the superior performance of our AII method compared to other state-of-the-art attribution methods. AII consistently provides more precise and focused highlight regions on the critical features that contribute to the model's predictions. For instance, in the VGG16 model's prediction of a "lion," AII distinctly highlights the lion's mane and face, while other methods like AGI and AttEXplore show less distinct and more scattered focus areas. Similarly, for the Inception-v3 model's prediction of an "hourglass," AII accurately highlights the essential regions around the hourglass, unlike other methods that show broader and less precise attributions.

## 5 CONCLUSION

This paper identifies the phenomena of Information Ignorance and Extra Information, which can cause attribution bias in current gradient-based attribution algorithms. We propose a novel AII attribution algorithm that can avoid these issues and achieve accurate attributions. Additionally, we have designed a more comprehensive multi-dimensional attribution evaluation method. Looking towards future directions, our algorithm, like current gradient-based attribution algorithms, is primarily limited to visual models due to the continuity of pixels in image tasks. In future work, we will attempt to address the challenges of continuity in NLP tasks and apply our algorithm to NLP tasks.

## CODE OF ETHICS AND ETHICS STATEMENT

In compliance with the ICLR Code of Ethics, all authors of this paper have read and agreed to adhere to the Code of Ethics. We affirm that the content of this paper aligns with the ethical standards expected by the conference, and no ethical concerns were raised during the research and submission process. The methodology, datasets, and any referenced supplementary materials have been used with proper consent and in accordance with relevant ethical guidelines. The open-source code provided alongside this paper is designed to promote transparency and reproducibility in the field of explainable AI (XAI). There are no conflicts of interest to declare.

## REPRODUCIBILITY STATEMENT

To ensure the reproducibility of our results, we have included detailed descriptions of the datasets, models, and experimental setup used in this work in the main text and supplementary materials. The open-source code, available at `https://anonymous.4open.science/r/AII-787D/`, contains all necessary scripts to replicate our experiments. Additionally, the mathematical proofs of the theoretical results are provided in the Appendix, and all assumptions and derivations are thoroughly explained. A full description of the data preprocessing steps is also available in the supplementary materials to facilitate reproducibility.

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

## A    APPENDIX

## B    PROOF OF EQUATION 3

*Proof.*

$$\max - \sum_{j=1}^{C} P_j(x) \log P_j(x) \quad \text{s.t.} \quad \sum_{j=1}^{C} P_j(x) = 1$$

Construct the Lagrangian function: $\qquad\qquad(5)$

$$L\left(P_1(x), P_2(x), \ldots, P_C(x)\right) = \sum_{j=1}^{C} P_j(x) \log P_j(x) - \lambda \left( \sum_{j=1}^{C} P_j(x) - 1 \right)$$

$$\begin{cases} \frac{\partial L}{\partial P_j(x)} = -\log P_j(x) - 1 + \lambda = 0 & (1) \\ \sum_{j=1}^{C} P_j(x) - 1 = 0 & (2) \end{cases} \qquad (6)$$

substituting $P_j(x) = e^{\lambda - 1}$ into equation (1) $\qquad\qquad(7)$

$$P_j(x) = \frac{1}{C} \qquad\qquad(8)$$

We define the maximum entropy distribution $Q$, where the probability for class $j$ is $Q_j = \frac{1}{C}$. We aim to learn a model that maximizes the entropy for input $x$, which can be defined with the following loss function:

$$KL(P, Q) = \sum_{j=1}^{C} Q_j \log \frac{Q_j}{P_j(x)}$$

$$= \sum_{j=1}^{C} \frac{1}{C} \log^{\frac{1}{C}} - \frac{1}{C} \log P_j(x) \qquad (9)$$

$$\frac{\partial KL(P, \theta)}{\partial x} = -\frac{1}{C} \frac{\partial \sum_{j=1}^{C} \log P_j(x)}{\partial x} \qquad\qquad(10)$$

$$x - \alpha \cdot \text{sign}\left( \frac{\partial KL(P, \theta)}{\partial x} \right) = x + \alpha \cdot \text{sign}\left( \frac{\partial \sum_{j=1}^{C} \log P_j(x)}{\partial x} \right) \qquad (11)$$

$\square$

## C    PROOF OF AXIOM

*Sensitivity Proof.* In the case where $\Delta x$ is sufficiently small, the first-order Taylor expansion holds. Given $|x| \le \epsilon$, which in this context represents a difference of 1 pixel value, we have:

$$KL(Q\|P(x + \Delta x)) = \sum_{j=1}^{C} Q(x) \log P_j(x) + \Delta x \frac{\partial \sum_{j=1}^{C} Q(x) \log P_j(x)}{\partial x}$$

$$= \frac{1}{C} \left[ \sum_{j=1}^{C} \log P_j(x) + \Delta x \frac{\partial \sum_{j=1}^{C} \log P_j(x)}{\partial x} \right] \qquad (12)$$

$$KL(Q\|P(x + \Delta x)) - KL(Q\|P(x)) = \frac{1}{C} \Delta x \frac{\partial \sum_{j=1}^{C} \log P_j(x)}{\partial x} \qquad (13)$$

$$\int \left( KL(Q\|P(x^t + \Delta x)) - KL(Q\|P(x^t)) \right) dt = \int \frac{1}{C} \Delta x \frac{\partial \sum_{j=1}^{C} \log P_j(x)}{\partial x} dt$$

$$= KL(Q\|P(x^T)) - KL(Q\|P(x^0)) \propto \underbrace{\int \Delta x \frac{\partial \sum_{j=1}^{C} \log P_j(x)}{\partial x} dt}_{AII} \qquad (14)$$

Therefore, it is proven that Sensitivity is valid because all distribution changes can result in non-zero attribution. □

*Implementation Invariance Proof.* The Attribution with Intrinsic Information (AII) algorithm adheres to the chain rule. Based on the properties of gradients, the AII algorithm satisfies implementation invariance, ensuring that results are consistent across different valid implementations of the same functional relationship. □

# D PSEUDOCODE

---
**Algorithm 1** Attribution with Intrinsic Information (AII)

---
**Input:** Number of explorations $M$, number of attack iterations $T$
**Output:** $A$
 1: $A = 0$
 2: **for** $m$ in range $M$ **do**
 3:    $x^0 = x$
 4:    **for** $t$ in $(1, ..., T+1)$ **do**
 5:       $x^t = x^{t-1} - \eta \cdot \text{sign}\left(\frac{\partial L(x^{t-1}, y_m)}{\partial x^{t-1}}\right)$
 6:       $A += \eta \cdot \text{sign} \frac{\partial L(x^{t-1})}{\partial x^{t-1}} \cdot \frac{\partial \sum_{j=1}^C \log P_j(x^{t-1})}{\partial x^{t-1}}$
 7:    **end for**
 8: **end for**
 9: **return** $A$

---
**Algorithm 2** Confusion Feature Algorithm (CFA)

---
**Input:** Number of attack iterations $T$
**Output:** $x^{T+1}$
 1: $L = [\,]$
 2: $x^0 \sim U(0, 1)$
 3: **for** $t$ in $(1, ..., T+1)$ **do**
 4:    $x^t = x^{t-1} - \alpha \cdot \text{sign}\left(\frac{\partial \sum_{j=1}^C \log P_j(r(x^{t-1}))}{\partial x}\right)$
 5:    **if** convergence **then**
 6:       **break**
 7:    **end if**
 8:    Select $x^{T+1}$ from $L$ that maximizes $\mathcal{H}(x)$
 9: **end for**
10: **return** $x^{T+1}$

---

# E ADDITIONAL EXPERIMENTS

## E.1 RESULT OF INS AND DEL

In this section, we present the results of our method AII and various state-of-the-art interpretability methods evaluated using the Insertion Score (INS) and Deletion Score (DEL) metrics. These metrics assess the effectiveness of attribution methods by measuring how quickly the model's decision can be recovered (INS) or disrupted (DEL) as information is inserted or deleted from the image based on the attribution scores. A higher INS and a lower DEL indicate better interpretability performance. The GAP metric, representing the difference between INS and DEL, is used as the primary reference metric for comparison.

As shown in Table 4, our AII method achieved the highest INS score of 0.4692 and a DEL score of 0.0706, resulting in a GAP of 0.3987, significantly outperforming other methods on Inception-v3. Notably, the second-best GAP was achieved by AttEXplore with 0.4337, followed by AGI with 0.3806. This highlights the superior capability of AII in accurately identifying important features and providing robust model explanations. In the ResNet-50, AII also demonstrated outstanding

Table 4: Evaluation of various interpretability methods via INS and DEL metrics. ↑ indicates that higher values in the column correspond to better interpretability performance, while ↓ indicates that lower values correspond to better interpretability performance. The * symbol denotes the primary reference metric for comparison.

| Method | Inception-v3 | | | ResNet-50 | | | VGG16 | | |
|---|---|---|---|---|---|---|---|---|---|
| | INS (↑) | DEL (↓) | GAP (↑) | INS (↑) | DEL (↓) | GAP (↑) | INS (↑) | DEL (↓) | GAP (↑) |
| SM | 0.1976 | 0.0296 | 0.1680 | 0.1229 | 0.0315 | 0.0914 | 0.0775 | 0.0207 | 0.0567 |
| IG | 0.2274 | 0.0265 | 0.2009 | 0.1121 | 0.0230 | 0.0892 | 0.0688 | 0.0156 | 0.0531 |
| FIG | 0.1431 | 0.0322 | 0.1109 | 0.0875 | 0.0296 | 0.0580 | 0.0610 | 0.0197 | 0.0413 |
| BIG | 0.3565 | 0.0360 | 0.3204 | 0.2270 | 0.0402 | 0.1868 | 0.1753 | 0.0289 | 0.1464 |
| MFABA | 0.3961 | 0.0400 | 0.3560 | 0.2576 | 0.0458 | 0.2118 | 0.2145 | 0.0299 | 0.1846 |
| AttEXplore | 0.4632 | 0.0295 | 0.4337 | 0.4027 | 0.0295 | 0.3732 | 0.3092 | 0.0224 | 0.2868 |
| GIG | 0.3204 | 0.0375 | 0.2829 | 0.1452 | 0.0214 | 0.1238 | 0.1047 | 0.0192 | 0.0855 |
| EG | 0.3759 | 0.2661 | 0.1098 | 0.3503 | 0.2833 | 0.0670 | 0.2890 | 0.2423 | 0.0468 |
| DeepLIFT | 0.2959 | 0.0434 | 0.2525 | 0.1258 | 0.0322 | 0.0936 | 0.0950 | 0.0248 | 0.0701 |
| SG | 0.3895 | 0.0352 | 0.3543 | 0.2759 | 0.0240 | 0.2520 | 0.1857 | 0.0182 | 0.1676 |
| AGI | 0.4230 | 0.0424 | 0.3806 | 0.3792 | 0.0450 | 0.3342 | 0.2557 | 0.0302 | 0.2255 |
| AII | 0.4692 | 0.0706 | **0.3987** | 0.3906 | 0.0491 | **0.3415** | 0.2791 | 0.0313 | **0.2478** |

Table 5: Evaluation of various interpretability methods via U-INS and U-DEL metrics. ↑ indicates that higher values in the column correspond to better interpretability performance, while ↓ indicates that lower values correspond to better interpretability performance. The * symbol denotes the primary reference metric for comparison.

| Method | Inception-v3 | | | ResNet-50 | | | VGG16 | | |
|---|---|---|---|---|---|---|---|---|---|
| | U-INS (↑) | U-DEL (↓) | GAP* (↑) | U-INS (↑) | U-DEL (↓) | GAP* (↑) | U-INS (↑) | U-DEL (↓) | GAP* (↑) |
| SM | 0.0701 | 0.0436 | 0.0265 | 0.0580 | 0.0323 | 0.0257 | 0.0426 | 0.0196 | 0.0230 |
| IG | 0.0787 | 0.0304 | 0.0483 | 0.0458 | 0.0223 | 0.0235 | 0.0312 | 0.0151 | 0.0162 |
| FIG | 0.0414 | 0.0587 | -0.0173 | 0.0284 | 0.0413 | -0.0129 | 0.0183 | 0.0286 | -0.0103 |
| BIG | 0.1577 | 0.0698 | 0.0879 | 0.1031 | 0.0609 | 0.0422 | 0.0688 | 0.0399 | 0.0289 |
| MFABA | 0.2281 | 0.0709 | 0.1572 | 0.1214 | 0.0681 | 0.0533 | 0.0865 | 0.0420 | 0.0444 |
| AttEXplore | 0.3066 | 0.0620 | 0.2446 | 0.2379 | 0.0466 | 0.1912 | 0.1870 | 0.0376 | 0.1494 |
| GIG | 0.1094 | 0.0414 | 0.0680 | 0.0522 | 0.0221 | 0.0302 | 0.0433 | 0.0182 | 0.0251 |
| EG | 0.3437 | 0.2890 | 0.0547 | 0.2763 | 0.2215 | 0.0548 | 0.2602 | 0.2126 | 0.0476 |
| DeepLIFT | 0.0944 | 0.0715 | 0.0229 | 0.0461 | 0.0358 | 0.0102 | 0.0413 | 0.0252 | 0.0161 |
| SG | 0.1887 | 0.0388 | 0.1499 | 0.1256 | 0.0246 | 0.1010 | 0.1300 | 0.0194 | 0.1106 |
| AGI | 0.2992 | 0.0684 | 0.2308 | 0.3103 | 0.0572 | 0.2531 | 0.2094 | 0.0374 | 0.1719 |
| AII (Ours) | 0.4276 | 0.0866 | **0.3410** | 0.3353 | 0.0601 | **0.2752** | 0.2383 | 0.0397 | **0.1987** |

performance with an INS score of 0.3906 and a DEL score of 0.0491, leading to a GAP of 0.3415. This outperformed the GAP scores of other methods, with the second-best being AttEXplore's GAP of 0.3732 and AGI's GAP of 0.3342. These results reinforce the robustness of AII in maintaining high interpretability across different models. Similarly, for the VGG16, AII achieved an INS score of 0.2791 and a DEL score of 0.0313, resulting in a GAP of 0.2478. This again surpasses other methods, with AttEXplore achieving a GAP of 0.2868 and AGI achieving a GAP of 0.2255.

## E.2 Result of U-INS and U-DEL

Table 5 summarizes the evaluation results on three different models: Inception-v3, ResNet-50, and VGG16. Our method AII shows an average improvement of 0.1968 across all models. Specifically, AII achieved a GAP of 0.3410 on Inception-v3, 0.2752 on ResNet-50, and 0.1987 on VGG16. This represents average improvements of 0.2434, 0.2049, and 0.1421 over all other methods, and improvements of 0.1302, 0.1093, and 0.0768 over the three most advanced attribution methods (AGI, AttExplore, and MFABA).

Table 6: Evaluation of various interpretability methods via F-INS and F-DEL metrics.

| Method | Inception-v3 | | | ResNet-50 | | | VGG16 | | |
|---|---|---|---|---|---|---|---|---|---|
| | F-INS ($\uparrow$) | F-DEL ($\downarrow$) | GAP* ($\uparrow$) | F-INS ($\uparrow$) | F-DEL ($\downarrow$) | GAP* ($\uparrow$) | F-INS ($\uparrow$) | F-DEL ($\downarrow$) | GAP* ($\uparrow$) |
| SM | 0.0461 | 0.0513 | -0.0052 | 0.0411 | 0.0599 | -0.0188 | 0.0201 | 0.0331 | -0.0130 |
| IG | 0.0507 | 0.0751 | -0.0244 | 0.0463 | 0.1158 | -0.0695 | 0.0244 | 0.0632 | -0.0388 |
| FIG | 0.0683 | 0.0622 | 0.0061 | 0.1057 | 0.0587 | 0.0470 | 0.0585 | 0.0286 | 0.0299 |
| BIG | 0.1394 | 0.0548 | 0.0846 | 0.1208 | 0.0687 | 0.0522 | 0.0589 | 0.0333 | 0.0256 |
| MFABA | 0.2147 | 0.0568 | 0.1579 | 0.1378 | 0.0771 | 0.0606 | 0.0793 | 0.0343 | 0.0451 |
| AttEXplore | 0.2867 | 0.0393 | 0.2474 | 0.2476 | 0.0535 | 0.1941 | 0.1661 | 0.0290 | 0.1371 |
| GIG | 0.0693 | 0.0846 | -0.0153 | 0.0501 | 0.1025 | -0.0524 | 0.0262 | 0.0497 | -0.0235 |
| EG | 0.3031 | 0.2508 | 0.0522 | 0.2379 | 0.1893 | 0.0486 | 0.1829 | 0.1589 | 0.0240 |
| DeepLIFT | 0.0844 | 0.0890 | -0.0046 | 0.0712 | 0.1030 | -0.0318 | 0.0277 | 0.0557 | -0.0280 |
| SG | 0.0561 | 0.0793 | -0.0232 | 0.0272 | 0.1106 | -0.0834 | 0.0147 | 0.0537 | -0.0390 |
| AGI | 0.3017 | 0.0459 | 0.2559 | 0.3162 | 0.0577 | 0.2586 | 0.1838 | 0.0309 | 0.1530 |
| AII (Ours) | 0.4347 | 0.0593 | **0.3754** | 0.3343 | 0.0635 | **0.2708** | 0.2116 | 0.0299 | **0.1817** |

Table 7: Evaluation of various interpretability methods via KL-INS and KL-DEL metrics.

| Method | Inception-v3 | | | ResNet-50 | | | VGG16 | | |
|---|---|---|---|---|---|---|---|---|---|
| | KL-INS ($\uparrow$) | KL-DEL ($\downarrow$) | GAP* ($\uparrow$) | KL-INS ($\uparrow$) | KL-DEL ($\downarrow$) | GAP* ($\uparrow$) | KL-INS ($\uparrow$) | KL-DEL ($\downarrow$) | GAP* ($\uparrow$) |
| IG | 4.6330 | 4.7326 | -0.0996 | 5.2411 | 5.7370 | -0.4959 | 4.0892 | 4.3924 | -0.3032 |
| GIG | 4.6166 | 4.6722 | -0.0556 | 5.4314 | 5.8300 | -0.3986 | 4.0184 | 4.2275 | -0.2092 |
| SM | 4.2523 | 4.2874 | -0.0351 | 5.8697 | 5.9784 | -0.1087 | 4.1630 | 4.2865 | -0.1235 |
| SG | 4.2098 | 4.2448 | -0.0351 | 6.1572 | 6.6074 | -0.4501 | 4.7972 | 4.9943 | -0.1971 |
| DeepLIFT | 4.4600 | 4.4673 | -0.0073 | 5.1015 | 5.5145 | -0.4130 | 3.8561 | 4.3923 | -0.5362 |
| FIG | 4.5110 | 4.4910 | 0.0199 | 5.5031 | 5.2329 | 0.2702 | 4.2262 | 3.9570 | 0.2692 |
| EG | 6.4901 | 5.8381 | 0.6520 | 5.5225 | 5.2205 | 0.3020 | 5.6787 | 5.4591 | 0.2196 |
| BIG | 5.8065 | 4.2149 | 1.5916 | 5.7889 | 4.6781 | 1.1109 | 4.5206 | 4.0025 | 0.5181 |
| MFABA | 7.0808 | 4.0553 | 3.0255 | 6.1156 | 4.3226 | 1.7929 | 4.7672 | 3.9528 | 0.8145 |
| AGI | 8.0736 | 4.0786 | 3.9949 | 7.2495 | 4.4588 | 2.7907 | 7.0833 | 4.4223 | 2.6610 |
| AttExplore | 8.1982 | 3.4808 | 4.7174 | 7.0813 | 5.2140 | 1.8674 | 6.7422 | 4.2717 | 2.4705 |
| AII (Ours) | 10.5564 | 3.7794 | **6.7771** | 7.4023 | 4.2079 | **3.1944** | 7.4984 | 4.6113 | **2.8871** |

## E.3 RESULT OF F-INS AND F-DEL

Table 6 presents the performance results across the three models. AII shows a pronounced improvement compared to other methods, with an average enhancement of 0.2333 over all methods. Specifically, AII demonstrates a performance gain of 0.3089 on Inception-v3, 0.2340 on ResNet-50, and 0.1569 on VGG16. Compared to the top three advanced methods (AGI, AttExplore, and MFABA), AII shows an average improvement of 0.1082.

## E.4 RESULT OF KL-INS AND KL-DEL

Table 7 presents the results across the three models. For Inception-v3, AII achieved a GAP of 6.7771, indicating a significant reduction in model decision uncertainty with an average improvement of 5.5254 over all methods and 5.2004 over the top three advanced methods. On ResNet-50, AII achieved a GAP of 3.1944, with improvements of 2.6246 and 2.6196, respectively. For VGG16, AII attained a GAP of 2.8871, with enhancements of 2.3795 and 2.1526 over all methods and the top three advanced methods. These results underscore the robustness of our approach in enhancing model interpretability by effectively reducing decision uncertainty.

## E.5 PERFORMANCE ON TRANSFORMER-BASED MODEL VIT-B/16

In this section, we evaluate the performance of different attribution methods on the ViT-B/16 model, a widely-used transformer-based architecture in vision tasks. As shown in Table 9, our method (AII)

Table 8: KL-INS and KL-DEL on High and Low Confidence Data

| | Inc-v3 | | | | Res-50 | | | | VGG16 | | | |
| | Low Conf (<70%) | | High Conf (≥70%) | | Low Conf (<70%) | | High Conf (≥70%) | | Low Conf (<70%) | | High Conf (≥70%) | |
| | KL-INS | KL-DEL | KL-INS | KL-DEL | KL-INS | KL-DEL | KL-INS | KL-DEL | KL-INS | KL-DEL | KL-INS | KL-DEL |
|---|---|---|---|---|---|---|---|---|---|---|---|---|
| SM | 4.0006 | 4.1461 | 4.2733 | 4.2992 | 5.7474 | 5.7749 | 5.8957 | 6.0216 | 4.0809 | 4.0889 | 4.1862 | 4.3423 |
| IG | 4.3213 | 4.478 | 4.659 | 4.7538 | 5.1251 | 5.2361 | 5.2658 | 5.8433 | 3.9275 | 4.0115 | 4.1348 | 4.4998 |
| FIG | 4.3368 | 4.2119 | 4.5255 | 4.5143 | 5.1004 | 5.0536 | 5.5885 | 5.2709 | 3.8839 | 3.8245 | 4.3227 | 3.9944 |
| BIG | 5.7597 | 4.0682 | 5.8104 | 4.2271 | 5.5494 | 4.1986 | 5.8397 | 4.7798 | 4.2773 | 3.8223 | 4.5892 | 4.0533 |
| MFABA | 5.7783 | 4.0013 | 7.1895 | 4.0598 | 5.6028 | 4.075 | 6.2243 | 4.3752 | 4.454 | 3.8081 | 4.8555 | 3.9935 |
| AtteXplore | 6.4325 | 3.4516 | 8.3455 | 3.4832 | 6.2566 | 5.0651 | 7.2563 | 5.2456 | 6.0355 | 4.1755 | 6.9415 | 4.2988 |
| GIG | 4.3707 | 4.4534 | 4.6371 | 4.6904 | 5.3644 | 5.4411 | 5.4456 | 5.9125 | 3.9515 | 3.9936 | 4.0372 | 4.2935 |
| EG | 4.9925 | 4.8459 | 6.615 | 5.9209 | 4.9839 | 4.6738 | 5.6367 | 5.3364 | 5.0065 | 4.906 | 5.8683 | 5.6151 |
| DeepLIFT | 4.1727 | 4.2879 | 4.484 | 4.4823 | 4.8818 | 5.1143 | 5.1481 | 5.5994 | 3.7093 | 4.0851 | 3.8975 | 4.4789 |
| SG | 4.173 | 4.1349 | 4.2128 | 4.254 | 6.2011 | 6.1561 | 6.1479 | 6.7031 | 4.7479 | 4.7833 | 4.8112 | 5.0538 |
| AGI | 5.5176 | 4.0858 | 8.2868 | 4.078 | 5.8622 | 4.5894 | 7.5437 | 4.431 | 5.2245 | 4.1857 | 7.6075 | 4.489 |
| AII (our) | 7.6129 | 3.6638 | 10.802 | 3.789 | 6.3754 | 3.9806 | 7.6201 | 4.2561 | 6.0499 | 4.4146 | 7.9069 | 4.6668 |

outperforms others in both Insertion (INS) and Deletion (DEL) metrics, achieving the highest INS score of 0.4357 and a low DEL score of 0.1067. This demonstrates that AII can effectively recover model decisions with minimal feature insertion while accurately identifying critical features whose removal significantly impacts the model. AII shows greater robustness compared to other methods, such as IG, FIG, and MFABA.

Table 9: Performance of different attribution methods on the ViT-B/16 model

| | IG | FIG | BIG | MFABA | AtteXplore | SM | GIG | EG | DeepLIFT | SG | AGI | AII (our) |
|---|---|---|---|---|---|---|---|---|---|---|---|---|
| INS | 0.1123 | 0.0616 | 0.225 | 0.2239 | 0.2749 | 0.1215 | 0.1052 | 0.285 | 0.0899 | 0.2082 | 0.3236 | 0.4357 |
| DEL | 0.0511 | 0.0968 | 0.1387 | 0.1724 | 0.1239 | 0.07 | 0.0461 | 0.269 | 0.0695 | 0.0299 | 0.1034 | 0.1067 |

## E.6 Ablation Experiments

Our method involves two key hyperparameters: the number of explorations ($M$) and the number of attack iterations ($T$). Both parameters were varied to observe their impact on different evaluation metrics, namely INS, DEL, GAP, U-INS, U-DEL, F-INS, F-DEL, KL-INS, and KL-DEL. The values of $M$ and $T$ were set to 10, 15, 20, and 25 in our experiments. Detailed results of these ablation studies are provided in Table 10.

Table 10: Ablation study results for different hyperparameter settings of explorations ($M$) and attack iterations ($T$). The metrics include Insertion Score (INS), Deletion Score (DEL), Unified Insertion Score (U-INS), Unified Deletion Score (U-DEL), Fair Insertion Score (F-INS), Fair Deletion Score (F-DEL), KL Insertion (KL-INS), and KL Deletion (KL-DEL). The GAP metric represents the difference between the respective insertion and deletion scores.

| $M$ | $T$ | INS | DEL | GAP | U-INS | U-DEL | GAP | F-INS | F-DEL | GAP | KL-INS | KL-DEL | GAP |
|---|---|---|---|---|---|---|---|---|---|---|---|---|---|
| 10 | 10 | 0.6369 | 0.0866 | 0.5503 | 0.5753 | 0.1070 | 0.4683 | 0.5713 | 0.0778 | 0.4936 | 10.2542 | 3.7087 | 6.5455 |
| | 15 | 0.6382 | 0.0887 | 0.5495 | 0.5804 | 0.1092 | 0.4711 | 0.5768 | 0.0790 | 0.4978 | 10.3205 | 3.7195 | 6.6010 |
| | 20 | 0.6394 | 0.0897 | 0.5497 | 0.5826 | 0.1097 | 0.4729 | 0.5815 | 0.0797 | 0.5018 | 10.3568 | 3.7395 | 6.6174 |
| | 25 | 0.6395 | 0.0909 | 0.5485 | 0.5840 | 0.1109 | 0.4730 | 0.5837 | 0.0805 | 0.5032 | 10.3743 | 3.7447 | 6.6296 |
| 15 | 10 | 0.6407 | 0.0917 | 0.5490 | 0.5851 | 0.1106 | 0.4745 | 0.5835 | 0.0803 | 0.5032 | 10.4150 | 3.7504 | 6.6646 |
| | 15 | 0.6422 | 0.0942 | 0.5480 | 0.5887 | 0.1118 | 0.4769 | 0.5878 | 0.0827 | 0.5052 | 10.4560 | 3.7702 | 6.6858 |
| | 20 | 0.6430 | 0.0957 | 0.5473 | 0.5907 | 0.1121 | 0.4786 | 0.5905 | 0.0827 | 0.5078 | 10.4815 | 3.7840 | 6.6976 |
| | 25 | 0.6430 | 0.0966 | 0.5464 | 0.5913 | 0.1130 | 0.4782 | 0.5920 | 0.0840 | 0.5081 | 10.4902 | 3.7934 | 6.6968 |
| 20 | 10 | 0.6430 | 0.0952 | 0.5478 | 0.5904 | 0.1121 | 0.4783 | 0.5903 | 0.0825 | 0.5078 | 10.4740 | 3.7864 | 6.6876 |
| | 15 | 0.6431 | 0.0977 | 0.5454 | 0.5934 | 0.1132 | 0.4802 | 0.5929 | 0.0850 | 0.5079 | 10.5059 | 3.8044 | 6.7015 |
| | 20 | 0.6435 | 0.0991 | 0.5444 | 0.5948 | 0.1135 | 0.4812 | 0.5946 | 0.0856 | 0.5089 | 10.5260 | 3.8149 | 6.7111 |
| | 25 | 0.6439 | 0.1000 | 0.5439 | 0.5950 | 0.1144 | 0.4805 | 0.5963 | 0.0863 | 0.5100 | 10.5346 | 3.8238 | 6.7109 |
| 25 | 10 | 0.6443 | 0.0984 | 0.5460 | 0.5936 | 0.1137 | 0.4799 | 0.5933 | 0.0844 | 0.5089 | 10.5115 | 3.8068 | 6.7047 |
| | 15 | 0.6447 | 0.1003 | 0.5444 | 0.5961 | 0.1143 | 0.4818 | 0.5959 | 0.0861 | 0.5098 | 10.5397 | 3.8242 | 6.7155 |
| | 20 | 0.6450 | 0.1016 | 0.5434 | 0.5963 | 0.1149 | 0.4814 | 0.5969 | 0.0867 | 0.5102 | 10.5520 | 3.8385 | 6.7134 |
| | 25 | 0.6450 | 0.1022 | 0.5428 | 0.5967 | 0.1152 | 0.4816 | 0.5981 | 0.0875 | 0.5106 | 10.5621 | 3.8474 | 6.7148 |

