# OpenReview forum: "Unbiased Attribution with Intrinsic Information"
_ICLR.cc/2025/Conference — Submitted to ICLR 2025_

### Official Review · Reviewer_kTtj · 2024-10-16

**Soundness:** 3
**Presentation:** 1
**Contribution:** 2
**Rating:** 3
**Confidence:** 4

**Summary:**

The paper proposes a new method to perform feature attribution called AII. They also suggest an approach (CFA) to find the correct null values for an image which would more closely emulate removal of superpixels from an image relative to a human. The AII algorithm they claim mitigates two issues other attribution methods have namely, feature ignorance and extra information.

**Strengths:**

+ Interesting setting
+ Reasonably thorough experiments

**Weaknesses:**

- Poor presentation
- No details given in the figure captions (1-5) of what models were used for the task and in some cases also the attribution method is missing. The text referring to them also does not contain this information.
- Presented solutions such as CFA are expensive
- Inconsistent and sometimes erroneous notation
- AII algorithm description extremely curt which is their main contribution
- No discussion or comparison with contrastive/counterfactual explanation methods

**Questions:**

The way the extra information part is written is confusing. The writing implies that somehow the including the class information during attribution is the extra information. What they mean by extra information only becomes clear when the discuss Figure 2, where it is more a statement about the input space and superfluous features selected therein.

AII algorithm description is quite short given that it is one of their main contributions. There is no discussion on why the 2 issues of extra information and information ignorance are mitigated by their approach. The approach essentially takes an average gradient over all class predictions, which also seems to have limited innovation (given that other averaging approaches such as Integrated Gradients etc. although somewhat different already exist).

Also the AII algorithm uses an adversarial attack strategy which is reminiscent of contrastive/counterfactual explanation methods (viz. contrastive explanations method (CEM), etc.). But no discussion is provided relative to those.

The CFA problem posed in Equation 2 is a hard optimization problem. To solve it for each image to find the null pixel values seems quite computationally intensive.

They consider just turning pixels black as the only solution for feature removal. However, that is not true. The idea in most of those methods is to insert a null value (also done in SHAP) which may be black pixels but could also be another value say average pixel value across the image or other images in the dataset. This phenomenon is not a new insight.

Equation 2, the sum uses index i but the term has j in it. $x_1, ..., x_n$ are not defined. Although I presume they imply pixel values.

$x^t$ I presume implies the $t^{th}$ image, but it is sampled from a univariate uniform distribution. If they are pixels then a subscript has become a superscript.

Experiments are reasonably thorough given that they compare against a bunch of different methods also using the setup of previous studies. However, I would have also liked to see timing results as the optimizations they propose for CFA and AII seem to be much more expensive than the attribution baselines.

Minor comment:
> "The larger the
attribution result, the more important that dimension is for the model’s decision."
I think this statement has to be qualified by saying larger the *absolute value* of the attribution ... Because a larger negative value also indicates an important feature


Overall, the paper requires significant rewriting in my opinion and is not currently ready for primetime.

---

> ### Author Response · Authors · 2024-11-19
>
> Since the Weaknesses and Questions raised are similar, we have primarily addressed Reviewer kTtj's questions.
>
> **Response to Question 1: Extra Information and Class Information**
>
> We appreciate the reviewer’s observation regarding the clarity of the "Extra Information" concept. In **Global Response 1**, we have provided a detailed explanation, including mathematical definitions, to clarify the concepts of "Information Ignorance" and "Extra Information."
>
> To summarize briefly:
> "Extra Information" occurs when attribution methods accumulate gradients under the assumption of a specific target class, leading to the inclusion of irrelevant features. In Fig.2, this manifests as attribution to superfluous regions, highlighting how predefined class assumptions can bias the results. This explanation will be expanded upon in the revised manuscript.
>
> ---
>
> **Response to Question 2: Description of the AII Algorithm**
>
> Thank you for highlighting the need for a more detailed discussion of how AII mitigates the issues of "Information Ignorance" and "Extra Information." Unlike existing methods, AII introduces adversarial attack strategies to remove the dependency on predefined target classes, a key source of "Extra Information." While averaging methods such as Integrated Gradients rely on class information, AII attributes features without such dependency, offering a novel contribution. Furthermore, we propose improvements to the commonly used Insertion and Deletion Scores to address their limitations, ensuring a more robust evaluation of attribution methods. Our experimental results consistently demonstrate the superior performance of AII across multiple metrics and dimensions.
>
> ---
>
> **Response to Question 3: Comparisons with CEM**
>
> We appreciate the reviewer's comment on the similarity between our method and contrastive explanation methods such as CEM. However, CEM does not satisfy the attribution axioms, whereas AII strictly adheres to them. Additionally, CEM relies on introducing samples similar to the current one, which risks adding "Extra Information." This makes it unclear whether the explanation pertains to the original sample or is influenced by the added samples. Our experiments ensure fairness by comparing methods that do not involve external sampling, demonstrating the effectiveness of AII. We will include a discussion on CEM in the revised paper for completeness.
>
> ---
>
> **Response to Question 4: Computational Cost of CFA**
>
> Thank you for your concern regarding the computational cost of CFA. In practice, the cost is not significant: CFA typically requires only 20 seconds per calculation and at most 90 seconds in extreme cases. Moreover, the null pixel values are computed once per model and reused, avoiding repeated calculations. This computational expense is negligible compared to the hundreds or thousands of hours required for model training.
>
> ---
>
> **Response to Question 5: Null Value for Feature Removal**
>
> While the use of black pixels or average pixel values as null values has been explored, our main contribution lies in addressing "Information Ignorance" and "Extra Information." Our entropy-maximization approach ensures that the chosen null value represents maximum uncertainty, aligning with mathematical definitions of entropy. This provides a theoretically grounded method for feature removal, which we believe is more robust than other heuristics.
>
> ---
>
> **Response to Question 6: Typographical Issues in Equation 2**
>
> Thank you for pointing out the errors in Equation 2. We have corrected the summation index from $i=1$ to $j=1$ and will provide more detailed descriptions for terms such as $x_1, \ldots, x_n$ and $x_t$. The iterative relationship for $x^0$ is already defined, ensuring the equation is complete. These revisions will be included in the updated manuscript.
>
> ---
>
> **Response to Question 7: Timing Results for AII and CFA**
>
> We agree that computational efficiency is important. To address this, we measured the FPS (frames per second) for AII and other baseline methods. The results are as follows:
>
> | Method      | FPS          |
> |-------------|--------------|
> | SM          | 66.52        |
> | IG          | 12.13        |
> | FIG         | 65.91        |
> | BIG         | 0.24         |
> | MFABA       | 29.28        |
> | AttExplore  | 0.29         |
> | GIG         | 287.64       |
> | DeepLIFT    | 5.43         |
> | SG          | 11.67        |
> | EG          | 41.46        |
> | AGI         | 0.14         |
> | **AII**     | **0.14**     |
>
> As shown, AII has comparable computational costs to AGI, while achieving significant performance improvements. Given that explainability prioritizes faithfulness and accuracy, we believe the additional computational expense is justified.
>
> ---
>
> **Response to Minor Comment: Absolute Value of Attribution**
>
> We agree with the reviewer’s suggestion and will revise the statement to clarify that larger absolute values of attributions indicate more important features for the model’s decision.

---

> > ### Author Response · Authors · 2024-12-01
> >
> > Dear Reviewer kTtj,
> >
> > Thank you for your detailed comments and observations, which have greatly contributed to improving our Submission 9383.
> >
> > In our rebuttal, we have addressed your concerns as follows:
> >
> > 1. **Extra Information Clarification**: We expanded on this concept in our global response, highlighting its connection to task-irrelevant features and provided examples to demonstrate its implications.
> > 2. **AII Algorithm**: We elaborated on its novelty and how it mitigates issues of "Information Ignorance" and "Extra Information" by leveraging intrinsic information without relying on target classes.
> > 3. **Comparison with Contrastive Methods**: We included a discussion on methods like CEM, explaining how AII adheres to attribution axioms and avoids reliance on external sampling.
> > 4. **Computational Cost**: We clarified that CFA computations are performed once per model and reused, with minimal impact on overall cost.
> > 5. **Timing Results**: We provided a comparison of frames-per-second (FPS) performance, demonstrating AII's comparable efficiency to AGI while achieving superior results.
> >
> > We hope these responses address your concerns. If you have any further questions, we would be happy to discuss them. We also humbly request you to reconsider your evaluation in light of our clarifications.
> >
> > Warm regards,
> > The Authors of Submission 9383

---

### Official Review · Reviewer_B4oD · 2024-11-03

**Soundness:** 3
**Presentation:** 2
**Contribution:** 3
**Rating:** 5
**Confidence:** 3

**Summary:**

This paper focused on feature-level data attribution and pointed out two possible issue in traditional gradient-based methods: Information Ignorance and Extra Information. The paper proposed Attribution with Intrinsic Information (AII), a new feature attribution method which accumulated the gradients for the sum of log predicted probabilities for all classes. Furthermore, the paper also proposed evaluation methods

**Strengths:**

- The paper did some evaluation to show that the method proposed (AII) outperformed other traditional methods.
- The proposed method empirically resolve the problem spotted by the paper (i.e., Information Ignorance and Extra Information)

**Weaknesses:**

- There should be a more intuitive and clear-to-understand illustration for Information Ignorance, Extra Information, what does the traditional algorithms care about and what does AII care about. It could be a diagram or a Venn diagram.
- It still hard to understand why AII could resolve the Extra Information (and the cause of extra information).
- More examples showing the problems (Information Ignorance, Extra Information) are resolved will be helpful.
- Some small typos
  - Table 2 caption: U-INS, U-DEL -> F-INS, F-DEL
- The experiment is carried out on image classification with some very popular dataset and models, it could be biased.

**Questions:**

- What’s the difference between high-confidence and low-confidence? Like high-confidence cover 90% and low-confidence cover 10%? Or it’s a 50-50 separate.
- Is there any possibility that the issues could be presented by math terms. Current illustration is clear (emperically) but not well-defined by anything other than plain text.

---

> ### Author Response · Authors · 2024-11-19
>
> **Response to Weakness 1: Illustration for Key Concepts**
>
> Thank you for the suggestion to provide a more intuitive and visual illustration of the key concepts. We have already provided detailed textual definitions of "Information Ignorance" and "Extra Information" in Section 1 (Introduction, lines 52-54) and Section 3.2 (lines 160 and 210). In brief, "Information Ignorance" refers to features that should be attributed but are omitted, while "Extra Information" involves attributing irrelevant features.
>
> To further enhance understanding, we will include diagrams (e.g., Venn diagrams) in a future revision to better illustrate the differences between traditional methods and AII in terms of what features they attribute.
>
> ---
>
> **Response to Weakness 2: Resolving Extra Information**
>
> We have elaborated on the causes and resolution of "Extra Information" in **Global Response 1**, including a mathematical formalization. In summary:
>
> - **Cause of Extra Information:** Traditional attribution methods accumulate gradients based on predefined target classes. This introduces assumptions about task-relevant features, leading to the inclusion of irrelevant information.
> - **How AII Resolves It:** AII eliminates the reliance on target class-specific gradients by redefining gradient accumulation to consider all class outputs, ensuring task-irrelevant features are excluded.
>
> We hope this explanation addresses the reviewer's concern.
>
> ---
>
> **Response to Weakness 3: Additional Examples**
>
> Figures 1–4 in the manuscript provide comprehensive examples of how our method addresses "Information Ignorance" and "Extra Information." Moreover, additional examples can be found in our supplementary materials at [https://anonymous.4open.science/r/AII-787D/rebuttal/]. These examples further demonstrate the effectiveness of AII in resolving these issues.
>
> ---
>
> **Response to Weakness 4: Typographical Errors**
>
> Thank you for pointing out the typographical error in the caption of Table 2. We have corrected "U-INS, U-DEL" to "F-INS, F-DEL" and conducted a thorough review of the manuscript to eliminate other potential typos.
>
> ---
>
> **Response to Weakness 5: Dataset and Model Bias**
>
> Our method is designed as a general attribution algorithm, and we chose well-known datasets and models to ensure transparency and reproducibility. Additionally, the baseline methods we compared against were also evaluated on these datasets, ensuring fairness in our experimental comparisons. The use of publicly available datasets and models is intended to provide a reliable and unbiased evaluation of our approach.
>
> ---
>
> **Response to Question 1: High-Confidence vs. Low-Confidence**
>
> High-confidence data refers to instances where the model is confident in its decision, implying fewer features interfere with the decision-making process. Low-confidence data, on the other hand, represents cases where the model's confidence is low, indicating that more features may be influencing the decision. This distinction is illustrated in Fig.1 of the manuscript.
>
> To address the question, we conducted additional experiments using a 50% confidence threshold as a dividing line. The results, summarized below, demonstrate that AII consistently achieves superior performance compared to other methods, even under this threshold:
>
> | Method      | Conf < 50% | Conf < 50% | Conf ≥ 50% | Conf ≥ 50% |
> |-------------|------------|------------|------------|------------|
> |             | INS        | DEL        | INS        | DEL        |
> | SM          | 0.0235     | 0.0179     | 0.0532     | 0.0618     |
> | SG          | 0.0250     | 0.0227     | 0.0396     | 0.0757     |
> | MFABA       | 0.0876     | 0.0264     | 0.2548     | 0.0664     |
> | IG          | 0.0306     | 0.0247     | 0.0570     | 0.0910     |
> | GIG         | 0.0284     | 0.0251     | 0.0597     | 0.0865     |
> | FIG         | 0.0236     | 0.0293     | 0.0824     | 0.0726     |
> | EG          | 0.1206     | 0.1324     | 0.3002     | 0.3153     |
> | DeepLIFT    | 0.0280     | 0.0266     | 0.0758     | 0.0854     |
> | BIG         | 0.0751     | 0.0275     | 0.1597     | 0.0635     |
> | AtteXplore  | 0.1193     | 0.0207     | 0.3395     | 0.0451     |
> | AGI         | 0.1145     | 0.0198     | 0.3609     | 0.0541     |
> | **AII**     | **0.2104** | **0.0226** | **0.5056** | **0.0709** |
>
> ---
>
> **Response to Question 2: Mathematical Formalization**
>
> We appreciate the suggestion to include more formal definitions of the issues. As discussed in **Global Response 1**, we have provided mathematical descriptions of "Information Ignorance" and "Extra Information." These definitions will be incorporated into the main text in a future revision to further clarify and strengthen the paper.

---

> > ### Author Response · Authors · 2024-12-01
> >
> > Dear Reviewer B4oD,
> >
> > We appreciate your thorough review and valuable suggestions on improving the clarity and presentation of our Submission 9383.
> >
> > In response to your feedback, we have made the following clarifications:
> >
> > 1. **Illustration of Key Concepts**: We provided detailed definitions and proposed including diagrams (e.g., Venn diagrams) in future revisions for better visualization.
> > 2. **Resolving Extra Information**: We explained how AII eliminates reliance on class-specific gradients, ensuring task-relevant features are accurately attributed.
> > 3. **Dataset Bias**: We clarified that widely-used datasets and models were chosen to ensure fair and reproducible comparisons.
> > 4. **Typographical Errors**: Errors in Table 2 were corrected, and the manuscript was thoroughly reviewed for other typos.
> >
> > If you have additional questions or suggestions, we are open to further discussions. We also kindly ask you to reevaluate our paper considering these updates.
> >
> > Warm regards,
> > The Authors of Submission 9383

---

> > > ### Comment · Reviewer_B4oD · 2024-12-02
> > >
> > > Thank authors for the reply. I acknowledge I have read all the reviews and rebuttal reply. I believe the experiments and the theory is acceptable and reasonable. I will adjust the score during the discussion period with other reviewers. The only concern is the writing, authors may consider a better presentation method in text rather than some images left for readers to imply the definition of some key concepts.

---

> > > > ### Author Response · Authors · 2024-12-02
> > > >
> > > > Dear Reviewer B4oD,
> > > >
> > > > Thank you for your recognition of our theoretical contributions and experimental results. We sincerely appreciate your thoughtful suggestions regarding the presentation of our key concepts. We assure you that in the final version of the manuscript, we will improve the textual descriptions and their integration with visual illustrations to ensure better clarity and readability.
> > > >
> > > > We humbly request that you reconsider the score during the discussion period, taking into account our commitment to addressing these concerns in the final version.
> > > >
> > > > Thank you once again for your constructive feedback and support.
> > > >
> > > > Best regards,
> > > > The Authors of Submission 9383

---

### Official Review · Reviewer_emWS · 2024-11-04

**Soundness:** 3
**Presentation:** 4
**Contribution:** 3
**Rating:** 5
**Confidence:** 3

**Summary:**

This paper introduces an approach named AII to address biased attributions given by existing approaches. The sources of biased attribution are categorized into information ignorance and recognition of irrelevant features. This work mainly proposes two concrete improvements: (1) more advanced feature removal through entropy maximization, which could be used to develop an unbiased evaluation metric, and (2) considering the information of all classes during attribution, which is used in AII.

**Strengths:**

The phenomenon of two attribution biases, including omission of important features and incorrect identification of irrelevant features as significant, is explained well with clear examples.

The extensive experiments show the great effectiveness of the proposed approach, primarily due to the high insertion score. This could be potentially explained from the provided examples as the explanations given by AII appear to be less noisy and concentrate more on important areas.

**Weaknesses:**

I am unsure about the settings of the example in Fig.2. This example tries to show that other algorithms wrongly capture the irrelevant square region. From the understanding of humans, the square region is indeed irrelevant. But the attribution methods are designed to explain model behaviours instead of human understanding, and the model might be erroneous and actually use that region for classification. That is, the example might also need to prove that, the model does not leverage that square region. Maybe it could be done by control experiments.

I like the interesting case presented in Fig.1, where the confidence is low due to ambiguity. However, it seems those cases could not be captured by the metrics used in experiments, because the calculating of scores does not involve reconstructing the exact same low-confidence prediction. Also it might be helpful to replace Fig. 6 or 7 with an example in such scenario.

I understand that the existing feature removal needs to be improved, but I don’t have a good intuition about why entropy maximization would help. Maybe more intuition could be added around lines 300-304.

I am concerned that the AII algorithm does not help explain less ambiguous figures such as Fig.3. And, for instance, in the MNIST data, it seems the large majority of figures are actually not that ambiguous.

**Questions:**

The observation in Fig.3 is interesting, and I wonder if this is a feature of the model or a “bug” in the attribution approach that needs to be addressed. If the same thing happens in the example of Fig.1, and considering the proposed AII algorithm is essentially summing all those attributions (maps), the attribution in Fig.1 should only highlight the dog area?

---

> ### Author Response · Authors · 2024-11-19
>
> **Response to Weakness 1: Fig.2 Settings and Irrelevant Regions**
>
> Thank you for pointing out the concern regarding Fig.2. The issue raised has been discussed in the work by Shah et al. [1], which highlights the phenomenon of "feature leakage" and how attribution methods may mistakenly highlight irrelevant features.
>
> In addition, the confidence score for this example is only 0.5786, while most examples in the MNIST dataset typically exhibit confidence scores above 0.95. Therefore, it is not appropriate to attribute this instance solely based on the current class, as traditional attribution methods require a predefined class to operate.
>
> Reference:
> [1] Shah, Harshay, Prateek Jain, and Praneeth Netrapalli. "Do input gradients highlight discriminative features?" Advances in Neural Information Processing Systems 34 (2021): 2046-2059.
>
> ---
>
> **Response to Weakness 2: Metrics and Low-Confidence Examples**
>
> We appreciate your suggestion to include more examples like Fig.1 in the paper. Additional examples demonstrating similar low-confidence scenarios are provided in our supplementary materials and can be accessed at [https://anonymous.4open.science/r/AII-787D/rebuttal/]. These examples further illustrate how our method effectively handles ambiguity in attribution.
>
> ---
>
> **Response to Weakness 3: Intuition Behind Entropy Maximization**
>
> Entropy represents uncertainty in a model's decision. Increasing entropy during feature replacement simulates a process where the model becomes progressively uncertain about its predictions. We believe this better approximates the removal of a feature compared to directly replacing it with a zero value, as zero replacement can inadvertently introduce biases depending on the model's learned features. This approach aligns better with the intuition that higher uncertainty indicates the effective removal of the feature's influence.
>
> ---
>
> **Response to Weakness 4: Less Ambiguous Figures and Fig.3**
>
> Regarding the concern about Fig.3, it is indeed based on the original MNIST resolution (32×32). The original image used in Fig.3 is available in our open-source code repository at [https://anonymous.4open.science/r/AII-787D/rebuttal/]. While the MNIST dataset predominantly consists of less ambiguous examples, Fig.3 demonstrates a broader limitation of existing attribution methods: reliance on predefined classes. Our method addresses this limitation by eliminating the need for such specifications, resulting in more robust and generalized attributions.
>
> ---
>
> **Response to Question 1: Observations in Fig.3 and Implications for Fig.1**
>
> The issue highlighted in Fig.3 reflects a limitation in current attribution methods, not a feature of the model itself. These methods assume that specifying a class will yield meaningful attribution for that class, which we demonstrate to be problematic. For instance, in Fig.3, the attribution results for a digit labeled "7" remain highly similar across all other class labels, suggesting that the process of class specification is redundant.
>
> Our AII algorithm avoids this limitation by performing attribution without requiring class specification, achieving superior results as shown in our experiments. In the context of Fig.1, it would not be reasonable to highlight only the dog area, as the confidence for the "dog" label is 0.53, with the remaining 0.47 confidence likely associated with the "cat." Simply focusing on the dog or cat alone would fail to explain why the "dog" confidence is only 0.53. Our method effectively addresses this issue by incorporating contributions from all relevant features, resulting in more faithful explanations.

---

> > ### Author Response · Authors · 2024-12-01
> >
> > Dear Reviewer emWS,
> >
> > Thank you for your detailed review and constructive feedback on our Submission 9383. Your comments on our examples and methods were very insightful.
> >
> > In response, we have addressed the following:
> >
> > 1. **Fig. 2 Example**: We clarified that the square region does not influence the model’s decision, as supported by control experiments and confidence scores.
> > 2. **Low-Confidence Cases**: Additional examples and explanations were included in our supplementary material to demonstrate AII's ability to handle ambiguity effectively.
> > 3. **Entropy Maximization Intuition**: We elaborated on how entropy maximization improves feature removal by representing maximum uncertainty, providing a theoretical basis for this approach.
> >
> > We hope these clarifications address your concerns. If there are any further questions or suggestions, we are happy to discuss them. We also kindly request you to reconsider your evaluation in light of these improvements.
> >
> > Warm regards,
> >
> > The Authors of Submission 9383

---

### Official Review · Reviewer_57hX · 2024-11-04

**Soundness:** 2
**Presentation:** 1
**Contribution:** 2
**Rating:** 3
**Confidence:** 4

**Summary:**

The paper introduces a novel feature attribution method, Attribution with Intrinsic Information (AII), designed to address challenges in current feature attribution algorithms, specifically regarding “ignored information” and “extra information.” AII accumulates gradients across the prediction vector, which may allow attribution map to capture features contributing to competing classes. Additionally, the paper proposes a a new evaluation metric that addresses a problem of the existing insertion and deletion metric. The effectiveness of AII is demonstrated through several experiments on image classification.

**Strengths:**

1. This paper presents a novel method for feature attribution.

2. The experiments include a variety of baseline feature attribution methods.

3. The code is available.

**Weaknesses:**

1. The key concepts, information ignorance and extra information, are not well-defined in the paper. Instead, these concepts are almost completely illustrated through examples in Figure 1 and Figure 2. While I appreciate the illustrative examples, it is unclear what exactly do they mean in a general setup.

2. The proposed method is described within half a page in Section 3.4. And it's unclear how does the proposed method mitigate the claimed problems (in fact, this may not be possible without clear definitions of the two problems).

3. Some claims do not seem accurate. For example, in line 157-158, I do not understand why "the loss function L(f (x), y) only contains the class information of y" since for cross-entropy loss, the loss also depends on the output logits for other classes.


Overall, the claimed problems are not well-defined, which undermines the motivation of the proposed method.

**Questions:**

See Weaknesses.

---

> ### Author Response · Authors · 2024-11-19
>
> **Response to Weakness 1 (W1): Definitions of Key Concepts**
>
> We acknowledge the reviewer's concern about the clarity of the definitions for "Information Ignorance" and "Extra Information." In response, we have provided detailed explanations in **Global Response 1**, where the definitions and mathematical formalizations of these concepts are presented. To summarize briefly:
>
> - **Information Ignorance** refers to the omission of features from non-target classes, leading to biased attribution.
> - **Extra Information** denotes the erroneous inclusion of irrelevant features in the attribution process.
>
> ---
>
> **Response to Weakness 2 (W2): Explanation of Mitigation**
>
> Our method addresses the problems of "Information Ignorance" and "Extra Information" by removing the dependency on a predefined target class during attribution. As highlighted in the manuscript, traditional methods rely on a specified class for attribution, which inherently excludes information from other classes, leading to "Information Ignorance." Additionally, if the chosen class is imperfect (e.g., has low confidence), the attribution process introduces "Extra Information," contaminating the results, as illustrated in Figure 1.
>
> The Attribution with Intrinsic Information (AII) algorithm avoids these issues by using a gradient accumulation method that considers the contributions of all class outputs without prioritizing a single class. This ensures a more balanced and unbiased attribution, as shown in our experimental results.
>
> ---
>
> **Response to Weakness 3 (W3): Clarification on Loss Function**
>
> The statement in lines 157-158 refers to the predominant focus of the loss function $ L(f(x), y) $ on the target class $ y $. While it is true that cross-entropy loss considers the logits for all classes, its primary optimization direction is to increase the output for the target class $ y $ while suppressing the outputs for all other classes. This can be understood from the gradient perspective: the gradient of the loss function with respect to $ f(x)_y $ increases $ f(x)_y $
>
> However, the gradients with respect to $ f(x)_{j \neq y} $ decrease those outputs. Thus, the optimization is inherently dominated by the target class $ y $, which can lead to "Information Ignorance" by neglecting contributions from other classes.
>
> We appreciate the reviewer's attention to detail and hope this explanation clarifies the intended meaning of our statement.

---

> > ### Author Response · Authors · 2024-12-01
> >
> > Dear Reviewer 57hX,
> >
> > Thank you for your insightful feedback on the Submission 9383, *"Unbiased Attribution with Intrinsic Information."* We greatly appreciate the time you took to review our work.
> >
> > We have carefully addressed your concerns, particularly:
> >
> > 1. **Clarity of Key Concepts**: We provided formal definitions and mathematical formalizations of "Information Ignorance" and "Extra Information" in our responses and clarified their distinctions.
> > 2. **Method Explanation**: We expanded our explanation of how AII mitigates the identified challenges by removing reliance on class-specific gradients and leveraging intrinsic information.
> > 3. **Loss Function Clarification**: We elaborated on the statement about the cross-entropy loss, highlighting its target-class prioritization as a source of bias.
> >
> > If you have any additional concerns or suggestions, we would be happy to discuss them further. Additionally, we kindly ask you to reconsider your evaluation of our paper in light of our detailed rebuttal and clarifications.
> >
> > Warm regards,
> >
> > The Authors of Submission 9383

---

### Author Response · Authors · 2024-11-19
**Global Response 1: Definitions of "Information Ignorance" and "Extra Information"**

We have provided clear and detailed definitions of "Information Ignorance" and "Extra Information" in Section 1 (Introduction, third paragraph, lines 52-54) and Section 3.2 (lines 160 and 210) of the manuscript. To briefly summarize:

- **Information Ignorance** refers to scenarios where features that should contribute to attribution are omitted.
- **Extra Information** denotes situations where features that are irrelevant to the task are mistakenly attributed as important.

For clarity, we restate the definitions here:

1. **Information Ignorance (Information Omission):**
   This phenomenon occurs when attribution methods fail to account for features from classes other than the target class, despite their potential influence on the model's decision-making process. For instance, when the target class is "dog," traditional attribution methods may completely ignore the presence of "cat" features in the image, even though the model considers both "dog" and "cat" features during decision-making. Such omissions can lead to biased attributions.

2. **Extra Information (Irrelevant Features):**
   This occurs when attribution methods mistakenly attribute irrelevant features to the task at hand. For example, an attribution method might highlight an unrelated region of an image as significant to the target class due to its reliance on specific loss functions, such as the maximum class output or cross-entropy, which introduce extraneous assumptions and biases.

We provide the following mathematical formalizations:

Let $a_i$ represent the true attribution score of a feature and $\tilde{a}_i$ denote the predicted attribution score. The set of important features is defined as $\Phi = \{i \mid a_i \geq \tau\}$, where $\tau$ is a threshold indicating activation intensity.

- **Information Ignorance:**
  Exists if there is a set $\varphi = \{i \mid i \in \Phi \text{ and } \tilde{a}_i < \tau\}$ with $|\varphi| \geq k$, where $k$ indicates the degree of feature omission. Larger $k$ implies more significant omission.

- **Extra Information:**
  Occurs if there is a set $\varphi = \{i \mid i \notin \Phi \text{ and } \tilde{a}_i \geq \tau\}$ with $|\varphi| \geq k$.

---

### Meta-Review · Area_Chair_YPRp · 2024-12-14

**Metareview:**

My recommendation is to reject the paper at this time. My decision stems from the consensus scores across reviewers and the lack of a champion. Given the lack of engagement, I also reviewed the submission personally – reading the reviews, responses, and the original submission. In this case, my recommendation is the same as those of the reviewers but for slightly different reasons. In particular, I recognize the importance of the problem but believe that it would benefit from further refinement and development to stand the test of time. What is missing in this case are ties to decision theory and validating feature attribution methods. Given this, I am including a list of papers below that could serve as inspiration to develop the work in the future.

- [Logic for Explainable AI](https://arxiv.org/abs/2305.05172) - presents a formal framework that could serve as a foundation for theory.
- [A Decision Theoretic Framework for Measuring AI Reliance](https://arxiv.org/abs/2401.15356)- presents a framework for how decision makers may be able to use side information to make better decisions.
- [Do Feature Attribution Methods Correctly Attribute Features?](https://openreview.net/forum?id=h4J41lQqaJ3) - includes some test cases (which could be used to test the validity of the current method)
- [Feature Responsiveness Scores: Model-Agnostic Explanations for Recourse](https://openreview.net/forum?id=wsWCVrH9dv) - highlights use cases of attribution on tabular datasets (which could be relevant here)

**Additional Comments On Reviewer Discussion:**

Three reviewers were only able to engage with the submission during the discussion period. Following the rebuttal and author-reviewer discussion, reviewer recommendations did not change substantially – and leaned toward rejection. Given the lack of engagement, I also reviewed the submission personally – reading the reviews, responses, and the original submission. In this case, my assessment is the same as those of the reviewers.

---

### Decision · Program_Chairs · 2025-01-22

Reject